# Recent Advances in Sandwich SERS Immunosensors for Cancer Detection

**DOI:** 10.3390/ijms23094740

**Published:** 2022-04-25

**Authors:** Aleksandra Pollap, Paweł Świt

**Affiliations:** 1Independent Researcher, 31-462 Krakow, Poland; aleksandra.pollap@gmail.com; 2Institute of Chemistry, Faculty of Science and Technology, University of Silesia in Katowice, 9 Szkolna Street, 40-006 Katowice, Poland

**Keywords:** cancer, biomarkers, surface-enhanced Raman spectroscopy, sandwich sensors, immunosensors

## Abstract

Cancer has been one of the most prevalent diseases around the world for many years. Its biomarkers are biological molecules found in the blood or other body fluids of people with cancer diseases. These biomarkers play a crucial role not only in the diagnosis of cancer diseases, but also in risk assessment, selection of treatment methods, and tracking its progress. Therefore, highly sensitive and selective detection and determination of cancer biomarkers are essential from the perspective of oncological diagnostics and planning the treatment process. Immunosensors are special types of biosensors that are based on the recognition of an analyte (antigen) by an antibody. Sandwich immunosensors apply two antibodies: a capture antibody and a detection antibody, with the antigen ‘sandwiched’ between them. Immunosensors’ advantages include not only high sensitivity and selectivity, but also flexible application and reusability. Surface-enhanced Raman spectroscopy, known also as the sensitive and selective method, uses the enhancement of light scattering by analyte molecules adsorbed on a nanostructured surface. The combination of immunosensors with the SERS technique further improves their analytical parameters. In this article, we followed the recent achievements in the field of sandwich SERS immunosensors for cancer biomarker detection and/or determination.

## 1. Introduction

Cancer has been classified as one of the major health problems worldwide for many years [1]. This disease is based on an uncontrolled overgrowth of cells and their division [2]. As a result, formed cancer tissue is composed of noncellular and cellular compartments that differ significantly in comparison with surrounding healthy tissue [3]. Although the term ‘cancer’ is commonly used interchangeably with the term ‘neoplasm’, in medical language, cancer is a specific type of neoplasm classified as a malignant neoplasm. The particular risk of this type of neoplasm lies in malignant cells, which can spread into other tissues and cause secondary tumors [2]. Cancer statistics for the year 2020, which were similar to those for previous years, are of great concern. Based on the data from the International Agency for Research on Cancer (IARC)’s GLOBOCAN, 19.3 million new cancer cases and almost 10.0 million fatal cases were estimated worldwide in 2020. Among the most prevalent diagnosed cancer types were female breast, lung, and prostate cancer (2.26, 2.21, and 1.41 million, respectively), while cancer deaths were mainly caused by lung, liver, and stomach cancer (1.79, 0.830, and 0.769 million, respectively) [4]. The causes of cancer are found in many differential factors, including genetic susceptibility through lifestyle (poor diet, tobacco smoking), exposure (radiation, electromagnetic fields), and environmental causes (residues of pesticides, industrial chemicals, air pollution) [5,6,7].

Many cases of cancer can be cured, especially when they are diagnosed early [8]. The development of rapid and sensitive methods of cancer detection remarkably enables faster treatment implementation and increases the chances of recovery [9]. Various techniques have been suggested for cancer detection. Standard techniques include X-rays, magnetic resonance imaging, and ultrasound scanning [10,11]. Currently, scientists’ interest is also focused on the investigation and development of new of imaging techniques based on microwaves, which, compared to the previously mentioned clinical techniques, are less costly and are based on harmless radiation [12].

The National Cancer Institute defines a biomarker as a biological molecule present in body fluids or tissues that signals normal or abnormal processes, conditions, or diseases [13]. Over the years, knowledge of cancer biomarkers (CB) has increased, enabling improvements in efficient detection and the efficacy of treatments due to their wide range of applications (Figure 1) [14].

The most popular biomarkers used in clinical practice are the prostate-specific antigen (PSA) for prostate cancer; the carcinoembryonic antigen (CEA) for gastrointestinal, breast, and lung cancer; CA125 for ovarian cancer; CA19-9 for pancreatic cancer; and CA15-3 for breast cancer [15].

Surface-enhanced Raman spectroscopy (SERS) is a compelling vibrational spectroscopy technique that was discovered in 1977 [16]. It is based on the enhancement of light scattering by analyte molecules when they are adsorbed on or near a nanostructured surface [17]. Currently, SERS is recognized as an attractive tool for (bio)chemical analytics when combining its molecular fingerprint specificity and single-molecule sensitivity [18]. The greatest advantages of this technique are its high sensitivity and specificity, nondestructive nature, and short analysis time [17,18]. These parameters are strongly related to SERS’ experimental considerations, such as substrate type, excitation source, and interaction between adsorbed molecules and the surface of plasmonic nanostructures [16].

Immunosensors are a group of biosensors that combine a biological recognition of antigens by an antibody due to an immunochemical reaction with a transducer part [19]. Specific antigen–antibody binding leads to highly sensitive and extremely selective detection of the analyte [20]. Great interest in the immunosensor issue is also related to their flexible application, ease of use, lower unit costs, automation, and reusability [21]. However, while ‘immunosensor’ and ‘immunoassay’ are similar terms, the differences between them should be clarified. While the immunosensor concept refers to antigen–antibody binding and recognition on the same platform, in the immunoassay system, the immunocomplex and biological identification occur in different places [20]. The most well-known immunoassay type is the enzyme-linked immunosorbent assay (ELISA) [21]. Direct ELISA applies an antibody labelled with an enzyme for attached solid-phase antigen detection [19]. To carry out an indirect ELISA strategy, two antibodies are required: a primary antibody binding the immobilized antigen and a secondary enzyme-linked antibody specific to the primary antibody [19,22].

In the sandwich ELISA, according to its name, the antigen is sandwiched between two antibodies, and one of them is labelled with an enzyme [20,22].

This article reviews the literature on the use of sandwich-type immunosensors in combination with surface-enhanced Raman spectroscopy (SERS) to detect cancer biomarkers. The paper aims to present the latest achievements in this research aspect. For this reason, it was decided to focus on the last 10 years, starting from 2011, and it was decided to take into account the newest publications from 2022 found in the Scopus, ScienceDirect, PubMed, Web of Science, and Google Scholar databases. During this period, almost 80 scientific publications on the subject of this review were published (Figure 2a). Until 2014, the number of articles each year did not exceed five publications. Since 2015, an approximately two-fold increase in the number of published scientific reports was observed. The most significant number of articles in a given year fell in 2019, when readers could become familiar with 12 new studies on the development of immunosensors for the detection of cancer biomarkers. This trend remained at a similar level until 2021. For this reason, it can be assumed that in the current year (2022), about 10 new approaches to this extremely interesting subject will be presented to the scientific community.

The described approaches focused mostly on research on model cancer antigens (CA), for which the appropriate immunocomplexes have been formed. Figure 2b shows the number of reports depending on a specific disease biomarker. The most frequently used biomarkers were: α-fetoprotein (AFP)—19 times, carcinoembryonic antigen (CEA)—21 times, and prostate-specific antigen (PSA)—17 times. Between two and five times, 11 cancer biomarkers were used and mentioned in publications individually or in multiplex analysis. The following antigens can be distinguished in this group: cancer antigens 125 and 19-9 (CA125 and CA 19-9, respectively), ferritin (FER), human epididymis protein 4 (HE4), interleukin-6 (IL-6), interleukin-8 (IL-8), mucin protein MUC4, neuron-specific enolase (NSE), squamous cell carcinoma antigen (SCCA), and vascular endothelial growth factor (VEGF). The remaining antigens were mentioned only once due to their lower prevalence. However, they are very diverse, and concern different types of neoplastic diseases. In the coming years, in addition to the most characteristic markers, the interest in new antigens will also increase. First of all, approaches will be developed to detect several antigens in a single analysis, allowing the detection of neoplastic disease at an even earlier stage.

## 2. Sandwich SERS Immunosensors Structure

Apart from the changes present as a result of the development of neoplastic disease, an important fact is that the concentration of some markers in the body fluids of patients may be a harbinger of a future neoplastic disease before the appearance of any symptoms. A specific reaction between an antigen and an antibody can be used to detect biomarkers and determine their level in body fluids; e.g., blood or serum. The most common test for this purpose is an enzyme immunoassay (ELISA), which uses enzymes to detect the reaction between the antigen and the antibody. In this case, an alternative based on the SERS technique and the phenomenon of immunoreaction is also up-and-coming.

### 2.1. Principle of the Sandwich SERS Immunosensor and Construction

The general principle of operation is similar to the ELISA test. However, SERS tags are used instead of the enzyme. A unique approach to this issue is the use of two metallic substrates instead of one, creating a sensor with a double amplification potential, the so-called sandwich assay sensor. The general scheme of the SERS immunosensor is as follows: metallic substrate–antibody–antigen–antibody–metallic nanoparticle. Double amplification of the signal occurs due to the metallic substrate’s interaction with the attached nanoparticles through a specific antigen–antibody interaction. A schematic representation of the SERS sandwich immunosensor is shown in Figure 3. This figure is presented in a multiplexed version, enabling the detection of three antigens representing different disease biomarkers. At the same time, the simplest and most frequently used version of the sensor relating to the analysis of a single antigen is marked (red box).

Basically, the antibody (capture antibody) is combined with the metallic substrate with an appropriate linker, most often with the -COOH group capable of binding the antigen. Samples are applied to the substrate functionalized in this way, and the antigen is then attached to the antibody. The most essential step is the attachment of metallic (gold or silver) nanoparticles (NPs) appropriately immobilized with antibodies (detection antibodies) and label particles (Raman reporters) to the antigen as a nanotag. Such a system can be modified to analyze more than one disease biomarker (multiplexed analysis). Antibodies can be attached to metallic substrates or nanoparticles via linkers or Raman reporters containing given functional groups. In the case of using two reporters (from both ends of the ‘sandwich’), the obtained spectrum is a combination of signals from both reporters, and the signal is accurate and true. This solution theoretically excludes the detection of a false positive signal.

Since SERS is a spectroscopic technique that measures the Raman scattering radiation of molecules adsorbed on the surface of a metal or a metallic sol particle, it results in a significant amplification of the measured radiation concerning the classical Raman measurement. In addition to the architecture of SERS substrates are the laser wavelength and molecule–metal interaction, which is the interaction of metallic substrates that enables signal amplification. The interaction of two metallic substrates with immunoassays allows for an additional increase in the intensity of the recorded signals, which is of particular importance in medical diagnostics as an ultrasensitive immunoassay, allowing the detection of even single molecules (with a limit of detection (LOD) at the fM level) [23].

### 2.2. Solid- and Liquid-Phase SERS-Active Substrate

Sandwich SERS-based immunoassays have been carried out on various platforms, which can be classified into two groups: solid substrates and liquid microbeads. Immunocomplexes in sandwich versions were usually created on solid surfaces, both metallic and nonmetallic SERS platforms, for biomolecular-interaction tests, separations of target analytes, and transduction of signals.

Immunoreactions can proceed much faster in liquid phases, which is associated with reducing the diffusion distance between the antigens and antibodies. Therefore, solid substrates were replaced with microbeads, both magnetic and nonmagnetic [23].

The first group consists of solid metallic substrates; it can also include various nonmetallic bases, but only if they have been modified with appropriate metallic layers, most often gold or silver. Several substrates were used to form sandwich immunosensors to detect and determine cancer biomarkers in this group. A few cases concerned the use of purely metallic substrates in the form of a gold nanoplate [24], Au plate [25], and gold array [26] for HE4, PSA, and AFP determinations, respectively. Bimetallic platforms involving the use of both gold and silver have also been used in the construction of immunosensors for cancer detection. This approach provided an additional amplification of the SERS signal. It is worth paying attention to the following suggestions in this area: nano-Ag/Au immune substrates [27,28], Au and Ag hexagonal nanoarray [9], and Au–Ag nanobox array substrates [29]. An extremely interesting approach involved the use of metallic electrodes as SERS substrates [30,31]. It is worth highlighting the version that relied on electrode-modified chitosan-stabilized AuNPs [32,33] and screen-printed electrodes [8].

A large group of proposed substrates includes glass, quartz, or silicon slides with a layer of nanoparticles, usually gold or silver, of various shapes (e.g., spherical, nanorods) [34,35,36,37,38,39,40,41]. To improve the intensity and repeatability of the SERS signal, substrates with a layer of silver or gold nanoparticles on microarray chips were also developed [42,43,44,45]; the most interesting case in this group was the use of ordered gold nanohoneycomb arrays [46]. The authors of various studies in the course of continuous development of and the search for new SERS substrates competed using increasingly original ideas. Wang and Lipert proposed the use of template-stripped gold substrates [47], while other authors used sandpaper or filter paper as a material on which metallic layers were applied [48,49,50,51]. However, these were not all proven substrates on which layers of metallic nanoparticles were applied, and among the most interesting cases used so far, it is worth mentioning NiCo_2_O_4_ nanorods [52], the hemisphere array [53], the polydopamine resin microsphere [54], the raspberry-like morphology of magnetic nanocomposites [55], the self-assembled monolayer of zwitterionic L-cysteine [56], delaminated Ti_3_C_2_T_x_ MXene [57], polymeric fibers [58], and the Fe_3_O_4_@TiO_2_ matrix [59].

The second group of solid SERS substrates used to form sandwich immunosensors was nonmetallic structures. This group accounted for a smaller number of cases than for solid metallic substrates. The most frequently used material in this group as a prospective candidate for SERS substrates was a molecularly imprinted polymer (MIP) in various versions, which were used in four different studies to determine AFP [60], CEA [61,62], and PSA [63]. Among other materials, polystyrene-based plates and nitrocellulose membranes were used. In the first case, polystyrene substrates were used for PSA analysis [64] and the simultaneous detection and determination of PSA and Rac [65]. However, the second material was used to determine AFP in two cases [66,67], and to simultaneously detect SCCA and CA 125 [68]. Quartz chips with punched wells [69], glass substrates [70], molybdenum disulfide [71], and polycarbonate filters [72] turned out to be less popular, but this does not preclude their future use.

The use of liquid microbeads as SERS substrates makes it much easier to carry out immunoreactions between antibodies and antigens, which significantly increases the speed of the entire analysis without a negative impact on the intensity of the SERS signal. The largest group of these substrates includes magnetic beads (MBs)/spheres in the basic version [73,74,75,76,77,78,79,80,81,82,83] or in various modifications: core-shell Fe_3_O_4_@AuNPs [84], Fe_2_O_3_@AuNP gold-coated microbeads [85,86], Au-coated NiFe magnetic NPs [87], γ-Fe_3_O_4_ microspheres [88], and boric-acid-functionalized magnetic silica particles [89]. Nonmagnetic particles were also used as potential microbead substrates. Gold/silver nanoshells were applied in two independent analyses of IL-6 [90] and in multiplex analysis of PSA, CEA, and CA 19-9 [91]. Among other applications, the use of 3D ordered silver nanoshell silica photonic crystal beads for CEA and AFP detection [92] or highly-branched gold nanoparticle substrates [93] is worth mentioning.

Using microbeads/spheres as SERS substrates is particularly useful when the analysis is performed with microfluidic or lab-on-a-chip systems.

### 2.3. Nanospherical SERS-Active Particles

SERS immunoprobes play an essential role in the detection and determination of cancer biomarkers. A SERS nanoprobe usually consists of several parts: metal NPs, Raman reporters, protection layers, and targeting molecules. However, not all of these aspects need to be used in the design. To record the SERS signal, it is necessary to attach Raman reporters to the surface of the metallic nanoparticles. Proper design of nanoprobes is required to obtain several desirable features, such as significant amplification and high stability of the SERS signal, avoiding loss of Raman reporters from the surface of NPs, sensitivity improvement, chemical and physical stability, targeting efficiency, and multiplexing ability [23].

As a detection element in the sandwich sensors developed so far, the most frequently used were gold [9,24,28,31,34,35,37,38,43,45,47,56,58,61,63,70,77,78,80,83,84,87,91,94,95] and silver [9,27,40,55,60,65,79,82,88,92] nanoparticles (AuNPs and AgNPs) with a shape similar to a sphere and of various sizes. These nanoparticles were each functionalized with a selected Raman reporter and a detection antibody to enable identification and determination of cancer biomarkers. SERS immunoprobes have been modified in order to generate sharp edges on their surface, thanks to which the recorded SERS signal was increased. Based on this assumption, attention was focused mainly on gold nanoparticles, leading their synthesis process in such a way as to obtain the desired structures in various forms: hollow gold nanospheres (HGNs) [26,42,73,75], Au nanostars (AuNSs) [8,44,46,62,69], Au rods [85], flowerlike AuNPs [86], and Au nanocages (GNCs) [54,93].

A separate group includes nanoparticles with a bimetallic structure. Gold–silver nanoshells were applied in the analysis of HER2 [30] and simultaneously analysis of SCCA and survivin [29]. In the case of tumor-derived exosome determination, gold-core–silver nanorods were used as the detection part [76]. This group includes silver particles (core) covered with a layer of gold (shell) [89] or gold particles (core) covered with a layer of silver (shell) [81]. One case also involved the use of -core–Au-shell nanoparticles [25]. Among the other shapes of bimetallic nanoparticles, nanocubes [71] and nanoshuttles [51] were also used.

To give the reader an idea of the number of modified nanoparticles proposed also with nonmetallic materials/substances, we decided to mention a few such cases. This group includes; e.g., nanotags with hybrid multilayered nanoshells prepared by the layer-by-layer assembly of small AgNPs at the surface of silica (SiO_2_) particles [53] also using poly(ethyleneimine) [74]; AuNP-coated acid-based resin microspheres [32]; silica-coated Ag nanorods [36]; core–shell SiO_2_@Ag [52]; SiO_2_-coated Si NPs [48]; AuNPs with polydopamine resin [33]; ZnO and CoFe_2_O_4_ nanocomplexes with Au [39]; AgNPs deposited on graphene oxide [64]; Ag-covered polystyrene spheres [41]; silica-coated gold/silver core-shell nanostars [66]; nanospheres with a silver coating core, an ultrathin continuous silica shell, and high coverage of gold nanospheres satellites [67]; MoS_2_ nanoflower@AuNPs [57]; Au-coated ‘stellate’ mesoporous SiO_2_@Au nanoprobes [49]; Au seeds on Fe_3_O_4_@TiO_2_ core–shell NPs [50]; nano-Ag polydopamine nanospheres [68]; and gold-graphene hybrid nanotags [72].

### 2.4. Preparation of Sandwich SERS Sensor

The preparation process for the SERS immunosensor in the sandwich version consists of several stages. The first step is the functionalization of the nanoparticles as a detection element. The next stage is based on the preparation of an active SERS substrate with attached antigens. Both these stages should be performed at the optimal time to carry out the last step of combining the prepared substrate with the functionalized nanoparticles. Figure 4 presents a generalized diagram showing the preparation of sandwich-type SERS immunosensors. This diagram was prepared based on various procedures described in the scientific literature, and is a generalization of various approaches [9,23,43].

Firstly, the detection immunoprobes are prepared (step 1). For this purpose, suitable metallic or other nanoparticles are functionalized with a selected Raman reporter. The reporter particles are bound to the surface of nanoparticles by the interaction of these nanoparticles in the reporter solution of a given concentration at a predetermined time. Next, the Raman reporter solution should be removed, and the nanoparticles should be rinsed in a suitable solution. It is very important to then attach linkers; i.e., substances that contain the -COOH group necessary for bioconjugation. Incubation of the particles with the linker (e.g., polyethylene glycol (PEG) or mercaptoundecanoic acid/mercaptoundecanol (MUA/MU)) at a selected concentration can take up to several hours. After washing the nanoparticles, appropriate compounds are added to activate the -COOH group located on the linker (if the Raman reporter used has such a group, there is no need to use an additional linker). Excess reagents are removed by centrifugation and resuspension in an appropriate solution; e.g., HEPES buffer. A solution of antibodies against a given antigen is added to a nanoparticle solution prepared in this way, and the entire solution is incubated for up to several hours. The detection antibodies are attached to the previously activated carboxyl group. Exemplary bovine serum albumin (BSA) solution should also be added to the nanoparticle solution to block active sites, which limits nonspecific binding. A selected solution with a small addition of a substance that blocks active sites is used for rinsing the nanoparticles. SERS-active nanoparticles prepared in this way are ready for the final stage of immunosensor preparation. In the case of multiplex analysis, the nanoparticles are prepared analogously with different reporters, so that the recorded SERS signals can be distinguished.

The second step is based on the preparation of functionalized substrates. Such a substrate of a given size is immersed in various solutions to attach subsequent elements. Then, after a given incubation time, it is washed in a suitable solvent, usually several times. First, the substrate is incubated with a linker (the most common version) or a Raman reporter (dual-tag system). For this purpose, suitable solutions are applied to the substrate, and the substances bind to the surface during an optimized time. After washing, the carboxylic group is activated and stabilized, analogous to the preparation of immunoprobes. Subsequently, a solution containing a capture antibody dedicated to a specific antigen is applied to the substrate, and the entire solution is incubated for up to several hours, after which the excess solution is removed by washing. In the case of multiplex analysis, a mixture of different antibodies is either applied or superimposed sequentially, one by one. The substrate is then immersed in the BSA solution to block nonspecific binding, the excess solution is removed, and the substrate is washed. Then, the antigen-containing solution or the test sample is applied to the substrate.

After binding of antigens (cancer disease biomarkers) and rinsing the substrate, in the final stage, functionalized SERS-active nanoparticles prepared in the first stage are applied and incubated. In this way, metal nanoparticles are conjugated with the capture substrate, forming a sandwich sensor. Before SERS measurements, such a system should be rinsed and dried. Information on the presence and concentrations of biomarkers are obtained based on the location and intensity of characteristic bands arising from Raman reporters.

## 3. Application of Sandwich SERS Immunosensors

### 3.1. Cancer Detection and Analytical Performances

Recent advances in sandwich SERS immunosensors for cancer detection are mainly focused on the determination of the prostate-specific antigen, α-fetoprotein, and carcinoembryonic antigen. Analytical performances applying the aforementioned tools are also aimed at the detection and quantification of squamous cell carcinoma antigen, mucin protein, interleukins (IL-6, IL-8), and other individual cancer biomarkers. However, while the vast majority of sandwich SERS immunosensors concern the one-component analysis, multiplexed approaches for the determination of two or more cancer biomarkers have also been developed. Here, we present sandwich SERS immunosensors for cancer detection while taking into account the biomarker type and several determined biomarkers.

#### 3.1.1. One-Component Analysis

##### PSA Detection

Prostate cancer is one of the most diagnosed cancer types in men [63]. The prostate-specific antigen, being a kind of kallikrein-like serine protease produced by the prostate gland, is recognized as a prostate cancer biomarker [64,78]. It plays a significant role not only in early diagnosis, but also in therapy. Thus, highly sensitive methods for its determination are desirable.

Hong et al. investigated a sandwich-type SERS immunoassay in which a microcontact printing method was applied for the patterned substrate [43]. The authors immobilized the captured antibodies on 5,5′-dithiobis (succinimidyl-2-nitrobenzoate)-coated gold with rhodamine 6G as a Raman reporter, conjugated the antigens on the layer of antibodies on AuNP layer on the patterned surface, and carried out capturing of antigens on the patterned surface. Selective conjugation of antibodies on the Raman reporter to the PSA enabled its detection at very low levels (1 pg mL^−1^). Zhao and coworkers proposed another method for the determination of the prostate-specific antigen [27]. The PSA was determined through the application of a sandwich-type nanostructure with AgNP immune probes and an AgNP immune substrate. It was found that the proposed method exhibited an extremely low LOD of 1.8 fg mL^−1^. Another SERS immunosensor for PSA detection was based on the immobilization of an anti-PSA antibody on 4-mercaptobenzoic acid molecules–Ag nanorods@SiO_2_ on a quartz slide coated with silver nanorods [36]. The LOD for the developed method was lower than that proposed by Zhao et al. [27], and was equal to 0.3 fg mL^−1^. Another work that supported SERS immunosensors applied in PSA determination was presented by Gao et al. [77]. The authors proposed a magnetic immunoassay technique using a microdroplet sensor, in which free and bound SERS tags were segregated by a magnetic bar embedded in a droplet-based microfluidic system. It was found that the LOD of the proposed immunoassay was below 0.1 ng mL^−1^, and the linear range was from 0.05 pg mL^−1^ to 200 ng mL^−1^. Functional core–shell nanoparticles (Au@1,4-benzenedithiol@AuNPs) formed the basis for another SERS immunosensor described in the literature [25]. Additionally, 1,4-benzenedithiol as a Raman reporter and core–shell hap marker were used. The proposed immunosensor showed a linear response toward increasing PSA in the range of 10 pg mL^−1^ to 10 ng mL^−1^, with a limit of detection of 2.0 pg mL^−1^. Moreover, the functioning of the proposed method was verified by the determination of PSA-spiked human serum samples with high accuracy (recoveries ranged from 92.0 to 96.5%). Xie and coworkers proposed another interesting method of PSA determination by applying a SERS immunosensor, in which a polyacrylamide-gel-contained zinc finger peptide was used as the ‘lock’, and zinc ions played the ‘key’ role [39]. The PSA was specifically connected with antibody-2-coupled ZnO nanocomplexes and an antibody-1-coupled magnetic nanocomposite. While Zn^2+^ formed from ZnO (under the HCl influence) destroyed the polyacrylamide-gel-contained zinc finger peptide, the structure of this peptide was opened, and the signal for toluidine blue (previously connected with an AgNP-coated silicon wafer) could be measured. The linear range and LOD of the developed sensor were 1 pg mL^−1^–10 ng mL^−1^ and 0.65 pg mL^−1^, respectively. The recovery studies based on PSA determination in spiked human blood serum showed a high accuracy due to recovery values close to 100% (98.1–102.4%). Yang’s group developed a SERS sensor based on enzymatic oxidation of the glucose substrate for H_2_O_2_ production. Its role was to dissolve Ag nanoparticles on a graphene oxide–AgNP composite by causing a reduction in the GO Raman signal [64]. The prostate-specific antigen was detected as a target by the formation of the sandwich immunoassay structure. The developed method exhibited a linear range of 0.5 pg mL^−1^ to 500 pg mL^−1^ and an LOD of 0.23 pg mL^−1^. The accuracy of the proposed immunoassay was verified by the determination of the PSA in human serum samples from clinically diagnosed patients. The results obtained by a chemiluminescence immunoassay as a reference method were in good agreement with the results found with the proposed approach. Gao et al. reported a promising SERS-based, pump-free microfluidic chip for PSA quantification [78]. The biomarker concentration was determined in the chamber where immunocomplexes were isolated by applying a permanent magnet. The linear range and LOD of the proposed immunosensor were found to be 0.01 ng mL^−1^–100 ng mL^−1^ and 0.3837 ng mL^−1^, respectively. Another example of PSA determination using a sandwich SERS immunosensor was proposed by Yun’s group [58]. The proposed method was carried out using AgNP-decorated electrospun fibers as the capture substrate, while the SERS signal was greatly amplified by the generation of hot spots between AgNPs on fibers and AuNP-based tags. The LOD of the proposed method was found to be equal to 1 pg mL^−1^, which was lower than that for approaches proposed by Gao [78], but higher than those described by Yang [64]. A SERS approach enabling PSA determination was also reported in Du’s work [50]. The immunoassay construction was based on Fe_3_O_4_@TiO_2_@Au nanocomposites as the immune probe and sandpaper coated with Ag as the immune substrate. The authors emphasized the recyclable properties of the developed immunoassay. It was found that less than 20% of the SERS intensity of the sandwich structure was lost after six cycles of immunoassay. An analytical investigation of the reported immunoassay exhibited a low LOD value (1.871 pg mL^−1^). Recently, Turan and coworkers combined a magnetic molecularly imprinted polymer and surface-enhanced Raman spectroscopy to develop a prostate biosensor [63]. The magnetic molecularly imprinted polymer was applied as an antibody-free capture probe. It was labelled by applying AuNPs modified with anti-PSA and 5,5′-dithiobis-(2-nitrobenzoic acid) as a Raman reporter. The LOD of the proposed immunosensor was found to be 0.9 pg mL^−1^, while the limit of quantification was equal to 3.2 pg mL^−1^. An analysis of the human serum as the real samples showed that the proposed immunosensor could be a useful tool for PSA determination. There was no significant difference between the biomarker determination by the proposed immunosensor and an ELISA test as a reference method. Moreover, the recovery values were close to 100%, reaching from 99.0 to 101.3%.

#### AFP Detection

The α-fetoprotein is a single-polypeptide-chain glycoprotein [34]. It was developed as a biomarker for the early diagnosis of liver cancer, known in the literature as hepatocellular carcinoma [38]. Additionally, AFP has also been rediscovered as a prognostic factor for hepatectomy, liver transplantation, and chemotherapy [66].

In a recently published work on a sandwich SERS immunoassay for α-fetoprotein by Lee et al. [26], the authors reported an automatic gold-array-embedded gradient microfluidic chip that integrated a gradient microfluidic device and gold-patterned microarray wells. In the proposed approach, the hollow gold nanospheres played the roles of SERS agents due to their high sensitivity and reproducibility, which was reflected in the analytical characteristics of sensor. The linear response toward the AFP concentration was obtained in a range of 0–10 ng mL^−1^, and the LOD was estimated at 0–1 ng mL^−1^. Wang and coworkers described another strategy for AFP determination [34]. Mercaptobenzoic-acid-labelled immunogold nanoparticles were combined with the antigen and the antibody atop. A linear relationship between the SERS signal and the AFP level was obtained, from 1 ng mL^−1^ to 100 ng mL^−1^, while the LOD was equal to 100 pg mL^−1^. Another sensor based on the boronate-affinity sandwich assay was developed by Ye et al. [60]. The reported assay relied on the sandwiches forming between boronate-affinity molecularly imprinted polymers, the target molecule, and boronate-affinity surface-enhanced Raman scattering probes. In this case, the linear range was wider than in Wang’s work [34], and ranged from 1 ng mL^−1^ to 10 µg mL^−1^. Moreover, the proposed assay was applied for AFP determination in the spiked serum samples. It was found that results obtained by the described tool were in good agreement with the reference method’s results. Thus, it was concluded that the developed method was accurate and could find a real application. A SiO_2_@Ag immune probe and a Ag-decorated NiCo_2_O_4_ nanorod immune substrate were proposed in another example of an immunoassay for α-fetoprotein determination [52]. For the immune probe preparation, Raman reporter molecules (4-mercaptobenzoic acid) and AFP were immobilized on SiO_2_@Ag in an application of magnetron-sputtering-enabled, Ag-decorated NiCo_2_O_4_ nanorods on carbon fiber cloth. The reported method exhibited a very low LOD, equal to 2.1 fg mL^−1^, and a wide linear range: 21 ng mL^−1^–2.1 fg mL^−1^. Ma and coworkers presented the following concept for an early hepatocellular carcinoma diagnosis [38]. The authors applied the frequency shift of 4-mercaptobenzoic acid for α-fetoprotein determination. Its basis was the interaction between AFP and anti-AFP on the Ag chips modified with 4-mercaptobenzoic acid. To carry out the sandwich immunoreaction on the chips, 5,5-dithiobis(succinimidyl-2-nitrobenzoate)-modified immunogold with AFP antibodies were involved. The immunosensor showed specific recognition of AFP in a linear range of 8 ng mL^−1^ to 1000 ng mL^−1^, with the corresponding LOD value equal to 0.5 ng mL^−1^. It was also confirmed that the proposed method could be applied in the clinical sample analysis of a male patient with hepatocellular carcinoma. Both the developed method and a chemiluminescence method (reference method) showed similar results. In a report prepared by Chen et al., another sandwich SERS immunosensor was described as a promising tool for AFP determination [96]. The authors developed a Ag/Fe/Ag sandwich cap-shaped SERS substrate for this purpose. The linear range and LOD of the proposed immunosensor were found to be 2 ng mL^−1^–8 ng mL^−1^ and 0.5 ng mL^−1^, respectively. Yang’s group also applied Au as a component of an immunosensor for hepatocellular carcinoma diagnosis [67]. The researchers developed Au@Ag@SiO_2_–AuNPs as the SERS-active core–shell–satellite nanostructures, which were modified with the α-fetoprotein antibody to obtain immune probes. For solid substrate preparation, nitrocellulose-membrane-stabilized captured anti-AFP antibodies were applied. The LOD was as low as 0.3 fg mL^−1^, while the linear response was recorded in the AFP concentration level from 1 fg mL^−1^ to 1 ng mL^−1^. The immunosensor functioning was examined toward α-fetoprotein determination in the spiked human serum samples. The analysis showed recovery values between 94.36% and 102.12%, suggesting a high accuracy of the developed method. An immunoassay reported by Zhao et al. was based on antibody immobilization on a nitrocellulose membrane for the AFP capture, and the antibody conjugated silica-coated gold/silver core–shell nanostars were the SERS probes [66]. The authors assessed the linear range (3 pg mL^−1^–3 mg mL^−1^) and the limit of detection (0.72 pg mL^−1^) toward the AFP. Zhe and coworkers presented a sandwich-type immune structure with Ag-covered polystyrene sphere probes and a Si@Ag substrate [41]. To prepare an immune probe, they immobilized 4-mercaptobenzoic acid and anti-AFP antibodies on the Ag-covered polystyrene sphere. The linear range was found to reach from 2 fg mL^−1^ to 200 ng mL^−1^, and the LOD was equal to 1.752 fg mL^−1^. Similar to the previously mentioned work of Ye [60], boronate affinity was also used by He et al. to develop a magnetic sandwich SERS immunosensor [89]. Its construction was based on the application of Ag@Au@4-mercaptobenzoic acid@anti-AFP as the detection probe, and Fe_3_O_4_@boric acid-functionalized SiO_2_@anti-AFP as a magnetic capture probe. The LOD of the proposed immunosensor reached 1.0 ng mL^−1^, and the linear response to the AFP concentration was 1.0 ng mL^−1^ to 1.0 mg mL^−1^. An analysis of the spiked human serum enabled an accurate assessment. As the recovery values were 85.8 to 105.7%, it was concluded that the developed method could be applied in AFP determination with high accuracy. Er and coworkers also reported a novel approach to α-fetoprotein determination [71]. For this purpose, the monoclonal antibody-modified MoS_2_ was applied as a capture probe, and the secondary antibody linked to rhodamine 6G was the Raman reporter. The proposed immunoassay exhibited a linear correlation in the following range: 1 pg mL^−1^–10 ng mL^−1^, with an LOD of 0.03 pg mL^−1^. To examine its functioning, the human samples were spiked with the biomarker. The recovery values reached 96.9 to 104.8%, showing a high accuracy of the developed immunoassay.

#### CEA Detection

The carcinoembryonic antigen is an acidic glycoprotein that is similar to the human embryonic antigen [61]. Determination of CEA concentration enables the diagnosis of colorectal cancer, breast cancer, ovarian carcinoma, and cervical carcinomas [33,94].

One of the methods for CEA determination was proposed by Guo et al. [75]. In the presented work, SERS tags were formed using hollow gold nanospheres embedded in 4-mercaptobenzonic acid, while magnetic microspheres were used as the substrates. A linear response was noticed in a concentration range between 10 pg mL^−1^ and 100 ng mL^−1^, while the LOD was equal to 10 pg mL^−1^. Surface-enhanced Raman scattering detection was also applied by Li et al. [87]. After bioconjugation, the probes, consisting of a Ni–Fe core and an Au shell, were magnetically focused on a spot in a microfluidic channel. As a result, the ‘hot spots’ were enriched, and SERS detection of the CEA was carried out. The described approach exhibited a limit of detection equal to 0.1 pg mL^−1^. Lin et al. investigated another method for carcinoembryonic antigen determination [94]. The authors prepared a Au@Raman reporter and γ-Fe_2_O_3_@Au, and modified both structures with a CEA antibody. In the presence of the carcinoembryonic antigen, the immunocomplex was received through the antibody–antigen–antibody. The linear response observed was in the range of 1 ng mL^−1^–50 ng mL^−1^, and the LOD was determined to be 0.1 ng mL^−1^. The proposed approach was tested during the analysis of spiked human serum. The recovery values reached 88.5% to 105.9%, showing the high accuracy of the developed method. Another immunoassay based on Nile blue labelling of polydopamine nanospheres was performed by Li et al. [33]. The developed sandwich structure, which consisted of anti-CEA/Nile blue/AuNP/polydopamine nanospheres, a carcinoembryonic antigen, and an anti-CEA/chitosan/AuNP/glassy carbon electrode were tested using both SERS and electrochemical detection of CEA. For the SERS-based immunoassay, a linear relationship was observed in the range of 2 ng mL^−1^–100 ng mL^−1^, while the lowest detected concentration was equal to 1.38 ng mL^−1^. It was noticed that in the electrochemical tests, the developed immunoassay exhibited better analytical parameters (the linear range and LOD were 1 pg mL^−1^–100 ng mL^−1^ and 0.68 pg mL^−1^, respectively). An antibody-free immunoassay was reported by Feng and coworkers [61]. The proposed approach was based on the application of 4-mercaptophenylboronic-acid-labelled AuNPs and boronate-affinity molecularly imprinted polymer spots on a glass slide. The limit of detection was found to be 0.1 ng mL^−1^. The practical application of the proposed method was also tested. Recovery values for CEA determination in the spiked serum samples from healthy people reached 78.5% and 80.0%, while in patients with liver cancer, the recovery values were found to be between 66.5% and 89.9%. Carneiro and coworkers developed another SERS-based method [62]. The authors applied a molecularly imprinted polymer incubated in a CEA sample, and finally in a SERS tag. For the SERS tag preparation, gold nanostars were linked to 4-aminothiophenol (Raman reporter) and an antibody for biomarker recognition. The LOD of the proposed approach was estimated at 1.0 ng mL^−1^. Medetalibeyoglu et al. reported another type of sandwich-type SERS immunosensor [57]. Its functioning involved 4-mercaptobenzoic-acid-labelled MoS_2_ nanoflowers@Au nanoparticles (SERS tag) and Ti_3_C_2_T_x_ MXene-functionalized Fe_3_O_4_@Au nanoparticles (magnetic substrate). The linear response toward the CEA concentration reached 0.0001 ng mL^−1^–100.0 ng mL^−1^, while the LOD was equal to 0.0001 ng mL^−1^. The recovery values for the spiked plasma sample analysis were close to 100% (98.37–100.04%), indicating the possibility of real application of the proposed immunosensor. A dual-detection approach that applied electrochemical and SERS techniques was reported by Castaño-Guerrero et al. [8]. The experimental part involved antibody binding on the Au-modified screen-printed electrode, labelling of a second antibody linked to gold nanostar-modified 4-aminothiophenol, and its incubation on the previously mentioned electrode surface. As an analytical parameter, the linear range (0.025 ng mL^−1^ to 250 ng mL^−1^) was determined. The same linear range was achieved by testing the SERS response in serum.

#### MUC4 and Interleukins Detection

Mucins are heavily glycosylated proteins that are recognized as biomarkers for pancreatic cancer [35,47].

To determine one of the mucins (MUC4), Wang and Lipert developed a SERS immunoassay [47]. In the proposed approach, the gold capture substrate was modified with an antibody, and the gold particles were modified with 4-nitrobenzenethiol and an antibody to form a sandwich structure. The linear range reached 0.01 μg mL^−1^ to 10 μg mL^−1^, and the limit of detection was equal to 33 ng mL^−1^. For verification purposes, the immunoassay was used for MUC4 detection in the serum samples. The authors highlighted that applying the conventional ELISA test was not possible, unlike the proposed method. Krasnoslobodtsev and coworkers proposed another approach to MUC4 detection [35]. The design of the reported immunoassay included gold modified with thiolated linker molecules and antibodies, as well as AuNPs functionalized with 4-nitrobenzenethiol and antibodies. The presented work showed the application of the proposed immunoassay for MUC4 detection in serum samples, and exhibited differences in SERS intensity for healthy people and patients with pancreatic cancer.

Cytokines are low-molecular-weight proteins that take part in different biological processes, including mediation or regulation of immune responses [90]. Their level in body fluids enables the diagnosis of diseases such as cancer [90,93].

Wang et al. proposed an approach to determine one of the cytokines—interleukin-6 (IL-6), relevant to, e.g., prostate cancer [90]. The authors employed gold/silver nanoshells (SERS labels) coated with a self-assembled monolayer of arylthiols (Raman reporter). The lowest detected concentration of IL-6 using the proposed method was equal to ca. 1 pg mL^−1^. Wiercigroch and coworkers developed a SERS-based sandwich immunoassay for the determination of the same interleukin [9]. In this investigation, silver and gold nanospheres were functionalized with 4-nitrotiophenol and α-mercapto-ω-amino PEG hydrochloride and linked to a detection antibody. The capture antibody was linked to a Ag/Au SERS substrate modified with 4-mercaptobenzoic acid. The linear response to IL-6 was 0 to 1000 pg mL^−1^, with an LOD of 25.2 pg mL^−1^. In this case, the limit of detection was higher than that of the method proposed by Wang [90]. A sandwich SERS immunoassay was also reported to determine that IL-8 was another form of cytokine [93]. To prepare the capturing substrates, highly-branched AuNPs were conjugated with anti-IL-8. To obtain SERS tags, gold nanocages were modified with 4-mercaptobenzoic acid and conjugated with anti-IL-8. The linear range reached 10 pg mL^−1^–1 mg mL^−1^, and the LOD was 6.04 pg mL^−1^. The analysis results for clinical serum samples when applying the proposed immunoassay were in agreement with the ELISA test due to the low relative errors.

#### Detection of Other Biomarkers

The squamous cell carcinoma antigen is a glycoprotein that exists in two forms: SCCA-1 and SCCA-2 [54]. It is commonly used as a biomarker for carcinoma of the uterus, cervix, lung, head, and neck [29].

For its determination, Lu et al. developed a surface-enhanced Raman scattering-based immunoassay [54]. Its structure was based on polydopamine resin microspheres coated with AuNPs and modified with an antibody (forming the capturing substrate), and hollow gold nanocages adsorbed on 4-mercaptobenzoic acid and modified with another antibody (SERS tag). To assess the analytical parameters of the reported immunoassay, the linear range (10 pg mL^−1^–1 mg mL^−1^) and limit of detection (8.03 pg mL^−1^) were determined. To verify the functioning of the proposed immunoassay, it was applied for SCCA determination in serum samples. The results of the analysis were in good agreement with the results obtained by the ELISA test.

Carbohydrate antigen 19-9 is known in medical analysis as the biomarker for colorectal and pancreatic cancer [48,59]. Tian’s group proposed a SERS-based immunoassay for its determination [59]. The authors assessed the linear range (1000 IU mL^−1^–0.001 IU mL^−1^) and LOD (5.65 × 10^−4^ IU mL^−1^) of the developed immunoassay. For verification purposes, the proposed immunoassay was tested as a tool for CA19-9 determination in serum samples. The relative error values between the examined immunoassay and chemiluminescence immunoassay were low, indicating the high accuracy of the proposed approach.

Li and coworkers reported a SERS immunosensor for determination of the vascular endothelial growth factor determination [44], which is recognized as a biomarker for tumor-associated angiogenesis. The authors immobilized the capture antibodies on a plasmonic gold triangle nanoarray pattern and the detection antibodies on gold nanostar@malachite green isothiocyanate@SiO_2_ nanoparticles. Analytical parameters such as the linear range (0.1 pg mL^−1^–10 ng mL^−1^) and LOD (7 fg mL^−1^) of the examined immunosensor were assessed toward the immunoglobulin G protein as a model. However, the developed tool was applied for vascular endothelial growth factor determination in blood plasma from patients with diagnosed breast cancer. The analysis results were in good agreement with the ELISA test.

Baniukevic et al. presented a SERS-based sandwich immunoassay as a useful tool for bovine leukemia virus antigen gp51 detection [85]. This virus is commonly known as an oncogenic retrovirus. For gp51 binding, magnetic gold nanoparticles were modified with antibodies, and gold nanorods coated with a 5-thio-nitrobenzoic acid layer with antibodies (Raman labels) were prepared. The linear response toward the analyte concentration and LOD were 0 mg mL^−1^–0.06 mg mL^−1^ and 0.95 mg mL^−1^, respectively. Moreover, the recovery studies carried out in the spiked milk samples showed a high accuracy of the method proposed (the recovery values were 85.5–100%).

The human epididymis protein 4 (HE4), a small molecule glycoprotein, is a common biomarker for ovarian cancer diagnosis, and on which Ge’s work was focused [84]. For its determination, a magnetic immunoassay was based on an antibody, 4-mercaptobenzoic-acid-coated AuNPs, and core–shell Fe_3_O_4_@Au nanoparticles modified with an antibody. The linear range was 1 pg mL^−1^ to 10 ng mL^−1^, and the LOD was equal to 100 fg mL^−1^.

To determine another cancer biomarker, such as the human epidermal growth factor receptor 2 (HER2), Wang et al. proposed an immunoassay using nanoscaled surface shear forces [33]. The immunoassay design included preparation of SiO_2_-encapsulated, fluorescence-integrated SERS Au/Ag nanoshells, conjugation of SERS particles with antibodies, and involvement of a microfluidic platform. It was demonstrated that the LOD was as low as 10 fg mL^−1^. The level of the aforementioned biomarker is very useful in cancer diagnosis, so the proposed approach was tested to compare the HER-2 levels in serum samples from healthy people and from HER2-positive patients.

Zong and coworkers proposed a SERS-based method for tumor-derived exosomes [76]. As tumor cells produce more exosomes with special biomarkers in comparison with the unchanged cells, they can be applied in cancer diagnosis. For preparation of the SERS probe, Au@Ag nanorods were covered with 5,50-dithiobis(2-nitrobenzoic acid) (SERS reporter), a SiO_2_ layer, and antibodies. To produce magnetic nanobeads, Fe_3_O_4_ nanoparticles were coated with SiO_2_ and antibodies. The smallest detectable number of exosomes using the proposed method was 1200 exosomes.

Boca et al. reported a sandwich SERS assay for metanephrine, which is a neuroendocrine tumor marker [37]. For label-free, nonspecific SERS detection, an Au nanoparticle/Au film structure was involved. The detection limit was equal to 0.197 µg mL^−1^ (10^−4^ M).

Bizzarri and coworkers developed an immunosensor for detection of the p53 tumor suppressor protein [70]. The concentration of this protein is highly correlated with the presence of breast cancer. The authors applied a biofunctionalized 4-aminothiophenol-AuNP complex with -N^+^≡N groups as the SERS probe and a glass substrate functionalized with aminopropyltriethoxy-silane and glutaraldehyde. The proposed method exhibited a linear range of 10^−10^ to 10^−17^ M. For verification purposes, the immunosensor was applied for p53 determination in spiked serum samples. The results obtained were in good agreement with the ELISA test results, showing the high accuracy of the proposed method.

A magnetic SERS immunosensor developed by Feng et al. was applied for human carboxylesterase 1 (hepatocellular carcinoma biomarker) determination [55]. To prepare the SERS tags, 4-mercaptobenzoic acid (4-MBA)-labelled AgNPs were involved, while for the SERS substrates, biofunctionalized Fe_3_O_4_@SiO_2_@AgNP magnetic nanocomposites were constructed. The linear range and detection limit were 0.1 ng mL^−1^ to 1.0 mg mL^−1^ and 0.1 ng mL^−1^, respectively. The SERS immunosensor was applied in the determination of human carboxylesterase 1 in the spiked serum samples. The recovery values were close to 100% (71.9–97.0%), showing the possibility of real application of the proposed approach.

A SERS-based immunosensor was also proposed by Panikar and coworkers [56]. In the presented work, the SERS substrate was an Au film modified with a self-assembled monolayer of zwitterionic L-cysteine and linked to an NKp30 receptor protein. The structure of the SERS nanoprobe was anti-B7-H6@adenosine 50-triphosphate@AuNPs. The immunosensor was applied in the determination of the B7 homolog 6 protein, a biomarker for cancer diagnosis. The linear response toward the biomarker reached 10^−10^ M to 10^−14^ M, with an LOD equal to 10^−14^ M (10.8 fg mL^−1^). For verification purposes, the immunosensor was used in the analysis of blood serum samples from cervical cancer patients. The results found were consistent with the ELISA test results.

Yang et al. proposed an immunoassay for determination of ferritin, a cagelike protein applied as liver cancer marker [49]. The immunoprobe was based on an Au-coated stellate SiO_2_, while the immunosubstrate involved Ag deposited on sandpaper assembled with filter paper and coated with antibodies. A linear relationship between the signal and marker concentration was found in a range between 1 × 10^−5^ g mL^−1^ and 3 × 10^−13^ g mL^−1^, and the LOD was 3.16 × 10^−14^ g mL^−1^.

Zhao’s group reported an immunoassay for quantification of MMP-9, which is one of the matrix metalloproteinases used for neoplasm diagnosis [88]. The authors applied AgNPs coated with 5,5′-dithiobis-(2-nitrobenzoic acid) and linked with antibodies and magnetic Fe_3_O_4_ particles conjugated with antibodies. To assess the analytical parameters of the proposed immunoassay, the linear range (0 pg mL^−1^ to 40 ng mL^−1^) and LOD (1 pg mL^−1^) were determined. Clinical application of the immunoassay was also tested. Results of analyses carried out using ELISA and the developed immunoassay were in good agreement.

The SERS-based platform proposed by Jibin et al. was developed for detecting circulating tumor cells [72]. For this purpose, AuNPs were modified with reduced graphene oxide and linked with antibodies, and the polycarbonate membrane was biofunctionalized. The limit of detection was equal to 5 tumor cells mL^−1^. To verify the possibility of real application, the proposed approach was tested in spiked blood samples. The recovery ranges were between 83% and 89.2%, indicating the high accuracy of the reported method.

Eom and coworkers presented a SERS immunoassay for determination of the human epididymis protein 4, which is recognized as an ovarian cancer biomarker [24]. To detect this biomarker, a capture antibody was linked to a single-crystalline gold nanoplate (immune substate) and an antibody immobilized on AuNPs (immune probe). The LOD was as low as 10^−17^ M.

#### 3.1.2. Two-Component Analysis

Here, we refer to SERS-based immunosensors involving the simultaneous determination of two cancer biomarkers.

Chon and coworkers proposed a novel approach to the simultaneous determination of CEA and AFP [73]. In its design, malachite green isothiocyanate (Raman reporter) and CEA antibodies were conjugated to hollow gold nanospheres and X-rhodamine-5-(and-6)-isothiocyanate (Raman reporter), and AFP antibodies were linked with HGNs. Additionally, magnetic beads coupled with anti-CEA and anti-AFP were prepared. The linear response toward the CEA and AFP concentrations were 0 to 100 ng mL^−1^ for both analytes. The immunoassay was also performed on serum samples, for which the linear range of CEA and AFP was narrower (0 ng mL^−1^–20 ng mL^−1^), and LODs were 1.67 ng mL^−1^ and 1.56 ng mL^−1^, respectively. Li et al. also proposed an approach to the simultaneous determination of the aforementioned biomarkers [92]. The authors reported silver nanoshell silica photonic crystal beads coupled with antibodies (SERS substrate), and 4-mercaptobenzoic acid/antibody/AgNPs as the SERS probe. The linear ranges for CEA and AFP were 0.01 pg mL^−1^–1000 ng mL^−1^ and 0.1 pg mL^−1^–1000 ng mL^−1^, respectively. The LOD for CEA was equal to 6.6 × 10^−6^ ng mL^−1^, while for APF, it was 7.2 × 10^−5^ ng mL^−1^. The possibility of immunoassay application was examined regarding CEA and AFP determination in clinical serum samples. The obtained results were consistent with those of the reference method, suggesting a high accuracy of the reported assay. The work of Gu et al. was also devoted to carcinoembryonic acid and α-fetoprotein determination [31]. Both markers were detected by combining electrochemical and SERS methods. The immunosensor preparation involved the preparation of a DNA-functionalized nanogold probe, construction of an electrochemical-based immunosensor by conjugation of antibodies, and finally a SERS-based immunosensor formed by adding AgNPs to the previously formed immune complex. The LOD values for CEA and AFP were 0.3 pg mL^−1^ and 0.6 pg mL^−1^, respectively. The linear range for carcinoembryonic acid was 5 pg mL^−1^ to 200 pg mL^−1^; while for the α-fetoprotein, it was 2 pg mL^−1^ to 100 pg mL^−1^. The recovery values obtained during the analysis of the spiked human serum samples were between 95.0% and 107.5%. Another approach to the simultaneous determination of CEA and AFP was presented by Li and coworkers [46]. Their SERS-based method was based on the application of Au nanostars (as SERS tags) and Au nanobowl arrays (as SERS substrates) conjugated with labelling and capturing antibodies. The proposed method was assessed using its linear range, which reached from 0.5 ng mL^−1^ to 100 ng mL^−1^ for both markers. The LOD values for CEA and AFP were as low as 0.41 ng mL^−1^ and 0.35 ng mL^−1^, respectively. The same parameters were evaluated by applying human serum. In this case, the linear range was 2 ng mL^−1^ to 100 ng mL^−1^ for both analytes. The LOD values were higher, and were equal to 0.44 ng mL^−1^ for CEA and 0.40 ng mL^−1^ for AFP.

Lee et al. reported a highly reproducible immunoassay for the determination of angiogenin (a protein involved in angiogenesis of tumor growth) and α-fetoprotein [42]. The researchers applied hollow gold nanospheres labelled with malachite green isothiocyanate (Raman reporter) and conjugated with an antibody. To prepare the SERS substrate, carboxyl-acid-modified gold wells conjugated with antibodies were used. The limits of detection values for angiogenin and α-fetoprotein were 0.1 pg mL^−1^ and 1.0 pg mL^−1^, respectively. Moreover, it was found that the dynamic range for the proposed immunoassay (10^−4^ g mL^−1^–10^−12^ g mL^−1^) was wider than that of the ELISA (10^−6^ g mL^−1^–10^−9^ g mL^−1^).

Wang and coworkers presented a SERS-based multiplex immunoassay for simultaneous determination of PSA and AFP [53]. For this purpose, a double SiO_2_@Ag (immune probes) and Au-film hemisphere array (immune substrates) was prepared. The developed immunoassay demonstrated a linear response toward PSA and AFP of 10 fg mL^−1^ to 400 ng mL^−1^. The LOD for PSA was as low as 3.38 fg mL^−1^; while for AFP, it was equal to 4.87 fg mL^−1^. The results of the serum sample analysis carried out using the reported method and the reference chemiluminescent immunoassay were consistent. It was concluded that the accuracy of the proposed method was high.

Lu et al. performed a multiplexing determination of a carcinoembryonic antigen and cytokeratin-19 [32]. Both biomarkers were applied for lung cancer diagnosis. The sandwich structure was based on Nile blue A-labelled AuNPs/aminosalicylic acid-based resin with an antibody and the a antibody linked to chitosan-stabilized AuNPs that were modified with a glassy carbon electrode. The LODs were 0.01 ng mL^−1^ and 0.04 ng mL^−1^ for CEA and cytokeratin-19, respectively. The linear relationship between the signals and the analyte concentrations were 0.05 ng mL^−1^ to 80 ng mL^−1^. The authors performed an analysis of serum samples by applying a sandwich SERS immunosensor and an ELISA test. The obtained results were similar, indicating the possibility of real application of the described approach.

Liang’s group reported an interesting approach involving an AgNP-based SERS–ELISA system [65]. The enzyme label of the ELISA enabled the controlled dissolution of 4-mercaptobenzoic acid (Raman reporter)-labelled AgNPs by applying H_2_O_2_. The proposed approach was used for the determination of PSA and ractopamine (an adrenal stimulant) in real samples. The LOD for PSA was as low as 10^−9^ ng mL^−1^. This analyte was successfully determined in human serum, with the results being in good agreement with time-resolved fluorescent immunoassays. The limit of detection for ractopamine was equal to 10^−6^ ng mL^−1^ in spiked urine samples.

Simultaneous detection of vascular endothelial growth factor and interleukin-8 was possible through the application of a surface-enhanced Raman scattering (SERS)-microfluidic droplet platform [82]. To develop this platform, Sun and coworkers conjugated antibodies to AgNPs and magnetic beads. The limits of detection for both cytokines were equal to 1.0 fg mL^−1^ in one droplet.

Xia et al. reported a SERS-based lateral flow immunoassay to detect the squamous cell carcinoma antigen and cancer antigen 125 (CA125), markers for a cervical cancer diagnosis [68]. To prepare two SERS immunoprobes, nitrocellulose membrane was modified with Raman reporters (4-aminothiophenol and 5,5-dithiobis-(2-nitrobenzoic acid)) and antibodies. The LOD values for SCCA and CA125 were equal to 7.156 pg mL^−1^ and 7.182 pg mL^−1^, respectively. The same parameter values determined in human serum were 8.093 pg mL^−1^ (for SCCA) and 7.370 pg mL^−1^ (for CA125). An analysis of clinical serum samples enabled an accurate assessment. The obtained results were in good agreement with the ELISA.

Lu and coworkers applied SERS spectroscopy in the simultaneous determination of the squamous cell carcinoma antigen and osteopontin (OPN), which is also recognized as a cervical cancer biomarker [51]. To prepare the immunoassay platform, Au–Ag nanoshuttles were applied as SERS tags, and Au nanoflowers were used as a capture substrate on hydrophobic filter paper. The limits of detection values in human serum samples were equal to 8.628 pg mL^−1^ (for SCCA) and 4.388 pg mL^−1^ (for OPA). An analysis of clinical serum samples using the SERS and ELISA methods showed that both methods led to very similar results.

An immunoassay proposed by Granger et al. was applied in serum carbohydrate antigen 19-9 (CA 19-9) and matrix metalloproteinase-7 (MMP-7) determination [45]. The first marker was described previously; MMP-7 is a protein whose expression is limited to the glandular epithelium and is related to early tumor development. The authors applied glass slides as SERS substrates, which were covered by Au and dithiobis(succinimidyl propionate) and coupled with antibodies. To prepare extrinsic Raman labels, AuNPs were modified with 5,5′-dithiobis(succinimidyl-2-nitrobenzoate) (Raman reporter) biofunctionalized with antibodies. The presented approach exhibited LOD values as low as 2.28 pg mL^−1^ and 34.5 pg mL^−1^ for MMP-7 and CA19-9, respectively. To verify the functioning of the proposed immunoassay, determinations of both biomarkers in human serum were carried out. The obtained results were consistent with the ELISA, indicating high accuracy of the reported approach.

Wang et al. described SERS-based dynamic monitoring in the simultaneous detection of two surface markers (CD19 and CD20) [79]. The authors prepared SERS probes consisting of Ag@4-mercaptobenzoic acid@SiO_2_ NPs and Ag@5,5-dithiobis(2-nitrobenzoic acid)@SiO_2_ NPs conjugated with antibodies. As a capture substrate, magnetic beads modified with SiO_2_ and functionalized with antibodies were applied. The estimated LOD in blood samples was one in one million cells (10^−6^). A blood sample analysis showed that the results obtained using proposed the method and flow cytometry were in good agreement.

Song and coworkers presented a SERS immunoassay for the determination of two lung cancer markers: the carcinoembryonic antigen and neuron-specific enolase (NSE) [86]. SERS tags were based on flowerlike AuNPs modified with 5,50-dithiobis-(2-nitrobenzoic acid) and antibodies, while the SERS substrate was prepared by coating magnetic nanoparticles with Au and conjugation with antibodies. The linear range was 1 ng mL^−1^ to 1 fg mL^−1^ for both analytes. The LOD values for CEA and NSE were 0.03 fg mL^−1^ and 0.66 fg mL^−1^, respectively. The lowest determined concentration for CEA in human serum was 1.48 pg mL^−1^, while for NSE, the same parameter in the same conditions was 2.04 pg mL^−1^. The linear range of biomarker concentration in human serum reached 10 pg mL^−1^ to 100 ng mL^−1^.

#### 3.1.3. Analysis of Three or More Components

Sandwich SERS immunosensors also have been applied in the determination of three or more cancer biomarkers.

Liu et al. developed an immunoassay for the determination of CEA, AFP, and CA 125 [74]. For this purpose, three thiol compounds: 3-methoxybenzenethiol, 2-methoxybenzenethiol, and 2-naphthalenethiol were used as SERS tags and biofunctionalized with antibodies (corresponding to CEA, AFP, and CA 125, respectively). The examination of immunoassay functioning was verified for biomarker determinations in spiked human serum samples. The concentration ratios of CEA, AFP, and CA 125 were at different levels. It was proved that the Raman signal intensities were independently changed toward an increase in the concentration of a given biomarker.

Li and coworkers reported an immunoassay for multiplexed detection of breast cancer antigens (CA 15-3 and CA 27-29) and the cancer embryonic antigen [69]. Immunoassay preparation involved gold nanostars modified with 4-nitrothiophenol as the Raman reporter and coated with SiO_2_ and antibodies corresponding to cancer antigens. Quartz SERS chips (SERS substrates) were also conjugated with antibodies. The linear response toward the concentrations of CA 15-3 and CA 27-29 was recorded between 0.1 U mL^−1^ and 500 U mL^−1^, while for CEA, the linear range was 0.1 ng mL^−1^ to 500 ng mL^−1^. The limits of detection were calculated to be 0.99 U mL^−1^, 0.13 U mL^−1^, and 0.05 ng mL^−1^ for CA 15-3, CA 27-29, and CEA, respectively. Results of the analysis of spiked fetal bovine serum showed that the proposed immunoassay enabled accurate determination of the aforementioned biomarkers due to high convergence with reference concentrations.

An immunoassay proposed by Zhou’s group was applied for prostate PSA, AFP, and CA 19-9 determination [48]. The sandwich structure consisted of nano-Si immune probes and a SiC@Ag SERS-active immune substrate. The detection limits of PSA, AFP, and CA 19-9 in human serum samples were as follows: 1.79 fg mL^−1^, 0.46 fg mL^−1^, and 1.3 × 10^−3^ U mL^−1^. To verify the real application, human serum samples were analyzed using the proposed immunoassay and a reference chemiluminescent immunoassay. The results obtained for both approaches were in good agreement.

Zhou et al. also proposed another SERS-based method for multiple cancer markers: PSA, prostate-specific membrane antigen (PSMA), and human kallikrein 2 (hK2) [40]. The authors used AgNPs as immune probes and SiC@Ag@Ag-NPs SERS as immune substrates. The LOD values were 0.46 fg mL^−1^, 1.05 fg mL^−1^, and 0.67 fg mL^−1^ for PSA, PSMA, and hK2, respectively. The linear range found for all biomarkers was 10^−5^ ng mL^−1^ to 10^1^ ng mL^−1^. An analysis of real samples such as clinical sera showed that the proposed method exhibited a high accuracy due to the high correlation between the results obtained using the developed approach and the reference chemiluminescent immunoassay.

Another procedure for three-component analysis was reported by Bai et al. [80]. The proposed immunoassay enabled simultaneous determination of three cancer biomarkers: AFP, CEA, and FER. The experimental section included preparation of poly-L-lysine-coated triple-bond coded AuNPs and their conjugation with antiAFP1, antiCEA1, and antiFER1, as well as the biofunctionalization of magnetic beads. The linear responses toward all biomarkers were in a range between 5 pg mL^−1^ and 1000 pg mL^−1^, with the LOD equal to 5 pg mL^−1^. Similar measurements were repeated in human serum samples. In this case, the linear ranges for AFP, CEA, and FER were: 0.5 pg mL^−1^–500 pg mL^−1^, 50 pg mL^−1^–2000 pg mL^−1^, and 10 pg mL^−1^–200 pg mL^−1^, respectively. The LOD values were: 0.15 pg mL^−1^ (for AFP), 20 pg mL^−1^ (for CEA), and 4 pg mL^−1^ (for FER). To verify the accuracy of the proposed approach, the clinical blood plasma samples were analyzed. The obtained results were in good agreement with the marker concentrations provided by a hospital.

Zhang and coworkers developed a sandwich immunoassay for simultaneous detection of three specific extracellular vesicle surface receptors: glypican-1, epithelial cell adhesion molecules (EpCAMs), and CD44 variant isoform 6 (CD44V6) [83]. To prepare SERS nanotags, AuNPs were functionalized with 5,5′-dithiobis(2-nitrobenzoic acid), 4-mercaptobenzoic acid, and 2,3,5,6-tetrafluoro-4-mercaptobenzonic acid (as Raman reporters) and antibodies. To form a sandwich structure, biofunctionalized magnetic beads were also applied. The LOD values were as low as 2.3 × 10^6^ particles mL^−1^. For verification purposes, the developed immunoassay was successfully used for determination of extracellular vesicle surface receptors in spiked plasma samples.

Karn-orachai performed another SERS-based approach being the base for three-component analysis [91]. The proposed method was used for PSA, CEA, and CA19-9 determination. The author proposed the application of two SERS-active materials (Au@Ag core–shell NPs) as the SERS substrate and mercaptobenzoic acid (MBA)-labelled AuNPs as the SERS probe. The work was focused on a correlation study between the polyelectrolyte nanodroplet and biomolecular size and the SERS signal enhancement.

Li and coworkers developed a SERS-based method for the determination of cytokines [81]. The authors prepared Au@MBA@AgNPs tunable SERS nanotags in which the MBA was 4-mercaptobenzoic acid, 5,5′-dithiobis (2-nitrobenzoic acid), and 2,3,5,6-tetrafluoro-4-mercaptobenzoic acid. The SERS nanotags were biofunctionalized with antigens. In the next step, magnetic beads were modified with protein G and conjugated with antibodies to form a complete immunoassay. For the multiplex detection of cytokines secreted from lymphoma, a stimulation with Con A (a protein factor that can induce the release of cytokines from the Raji cell line) was necessary. While no INF-γ, TNF-α, or IL-10 were detected without Con A stimulation, after this process, concentration values of 132.63 pg mL^−1^, 95.51 pg mL^−1^, and 156.65 pg mL^−1^ (for INF-γ, TNF-α, and IL-10, respectively) were found.

Kamińska and co-workers reported a SERS-based immunoassay in a microfluidic system for multiplexed detection of interleukins: IL-6, IL-8, and IL-18 [28]. The authors applied AuNPs coated with Raman reporters (5,5′-dithio-bis(2-nitro-benzoic acid), fuchsin, and p-mercaptobenzoic acid)) and conjugated with corresponding antibodies. To form a sandwich structure in the presence of biomarkers, the next layer consisted of an Ag–Au surface modified with 6-amino-1-hexanethiol and anti-IL-6, anti-IL-8, and anti-IL-18. The LOD values in a simultaneous multiplexed method applied in testing blood plasma samples were: 3.8 pg mL^−1^, 7.5 pg mL^−1^, and 5.2 pg mL^−1^ for IL-6, IL-8, and IL-18, respectively.

Xue et al. proposed another SERS immunoassay for multiplex detection of PSA, AFP, CEA, and the neuronal surface antigen (NSA) [95]. The proposed method involved immobilization of antibodies on a gold surface using photoirradiation, as well as modification of AuNPs with a Raman reporter (4-mercaptobenzoic acid), linker (polyethylene glycol), and labelling with antibodies. It was found that the LOD of the proposed assay could be as low as 10^−12^ mol mL^−1^ for all four cancer biomarkers.

### 3.2. Sample Preparations

The analysis of real samples enables the assessment of the accuracy of the proposed method. One of the most common types of samples analyzed in the detection of cancer biomarkers is blood. When applying a sandwich SERS immunosensor, biomarkers could be detected in whole blood, serum, or plasma [25,54,57].

Whole blood without any pretreatment was applied, e.g., for SCCA, MMP-9, CTCs determination [54,72,88]. Centrifugation of blood was carried out for MUC4 and IL-8 determination [35,93].

In preparation of serum samples, they are often centrifugated and/or diluted with phosphate buffer solution (PBS) to decrease the sample viscosity [25,31,45,48,53,67,73,74]. This approach was applied, e.g., for PSA, AFP, or simultaneous CEA and AFP determination [25,67,73]. Application of serum samples without any purification and modification was also found in the literature [38,64,68,71,94]. In addition to serum, plasma has been tested as well [28,57]. It may be prepared in a similar way to serum: by centrifugation and dilution with PBS [57]. This method of real sample preparation was applied in CEA determination [57].

An example of another substance used as a real sample is milk, in which oncogenic retroviruses can be present. One of them is the bovine leukemia virus antigen gp51, which was determined in milk without any special pretreatment [85].

## 4. Detection Methods

The construction of sandwich-type SERS immunosensors has been subjected to different modifications in the course of scientific research, leading to the implementation of various analytical goals. One of the solutions introduced to the basic version of immunosensors is the possibility of performing multicomponent/multiplexed analysis for at least two disease biomarkers. Another approach involves the use of a double-tag system on both the substrate and the attached nanoparticles, which allows the elimination of false-positive signals. On this basis, the proposed immunosensors can be divided into several groups: single-tag systems, dual-tag systems, and one component and multiplexed analyses. Table 1 lists different versions of SERS sandwich immunosensors and the group in which each approach can be allocated.

### 4.1. Single Raman Reporters (RR)

The most frequently used systems were sensors with a configuration containing only a detection element—nanoparticles (functionalized with a Raman reporter) attached to the antigen. Such systems were usually built in the following configuration: substrate–antibody–antigen–antibody–nanoparticles. After creating such a sensor, the analytical signal came from a Raman reporter coupled with metallic nanoparticles. Based on the characteristic bands of the Raman spectrum specific to the reporter used, it is possible to perform both qualitative analyses to confirm the presence of a given biomarker and a quantitative analysis that allows for determining the concentration of biomarkers in the tested sample. A single signal from a specific band facilitates the interpretative abilities and enables data analysis to be significantly faster and easier. The disadvantage of this approach is the inability to eliminate false signals that may result not from specific attachment to the antigen, but from the aggregation of nanoparticles on the surface of the metallic substrate, or other nonspecific interactions.

About 95% of the presented studies on SERS immunocomplexes of the sandwich type for the detection of cancer biomarkers can be classified into this group. The use of only one label on the detection element in the context of a given antigen is the simplest form of the immunosensors under consideration. The complexes prepared in this way were applied to various antigens, using all the Raman reporters mentioned in Table 1. In the case of using only one detection element, the analysis could be carried out in both the single- and the multiplex analysis versions. The various analyzed antigens, along with the Raman reporters provided, and basic information on the structure and basic validation parameters of the proposed sensors are included in Table 1.

### 4.2. Dual-Tag Systems

The problem of false-positive signals was solved by double-tag application during the construction of sandwich sensors. False-positive signals may result from the aggregation of metallic particles before adding them to the functionalized substrate or directly on the surface of the SERS platform. In this situation, ‘hot spots’ are generated between the edges of nanoparticles, but not as a result of the interaction of nanoparticles with substrates containing appropriate metallic nanostructures on the surface. Despite the strong analytical signals obtained in this manner, this effect is unfavorable due to the presence of false signals. The entire analysis then becomes unreliable and erroneous. The second source of false-positive results is defects on the metal platform surface, such as very small clefts. In such spaces, nanoparticles can accumulate, giving amplified Raman signals even though the sensors are not formed as a result of the connection between the antibody and antigen [97].

This problem can be solved by modifying the immunosensor by attaching an additional Raman reporter tag to the SERS nanostructured platform. In this way, a double-label system (dual-tag) is created, with one tag on the substrate and the other on the nanoparticles. After measuring the analytical signals, an appropriate data conversion should be performed to extract information relevant to the qualitative and quantitative analyses. The bands characteristic of both used Raman reporters are taken into account. The signal included in the qualitative and quantitative analysis is most often the ratio of the intensity/area measured for both characteristic bands of both reporters. The distinction between true and false-positive signals is made based on the chemometric analysis of the results obtained [9,97].

A dual-tag system was proposed in several studies. The best example of this type of immunocomplexes was used to determine interleukin-6 using different combinations of metallic platforms and nanoparticles: Au–Ag, Au–Au, and Ag–Ag. The best enhanced SERS signal was generated by an Au substrate in combination with AuNPs. A dual-tag paradigm was realized by attaching the Raman reporter to the substrate (mercaptobenzoic acid (MBA)) and the second one to the nanoparticles (4-nitrothiophenol (NTP)). The correct formation of the sandwich form was represented by the presence of characteristic bands in the Raman spectra. The band at 1338 cm^−1^ was related to the NTP present on the nanoparticles, while the second band considered in the analysis was the combination of the two bands at 1572 cm^−1^ (NTP), and the third was associated with MBA present on the metallic substrate at 1588 cm^−1^. The analytical signal was the ratio of the two mentioned values, which made it possible to compensate for false-positive signals and the effects of nonspecific binding of antibodies and antigens. These studies made it possible to obtain a wide linear range (0 pg mL^−1^–1000 pg mL^−1^) with a good sensitivity (LOD of 5.2 pg mL^−1^) and with good precision (RSD < 10%) [9].

A similar concept was developed for the determination of AFP and AFP-L3 [38]. MBA-labelled silicon chips coated with Ag were used to determine AFP. In the next step, DSNB-modified AuNPs were functionalized with AFP-L3 antibodies for sandwich reaction on the silicon chips. An exponential linear relationship with AFP-L3 concentrations, excellent reproducibility, and a high accuracy were obtained based on the experiments. Additionally, due to the use of two SERS tags, it was possible to evaluate frequency-shift and intensity-monotonic changes.

Another dual-tag system was applied to detect and determine the squamous cell carcinoma antigen and osteopontin in cervical cancer serum [51]. Au–Ag nanoshuttles were functionalized with MBA and 5,5′-dithiobis(2-nitrobenzoic acid) as detection tags and combined with capture parts—hydrophobic filter-paper-based Au nanoflowers modified with dimercaptosuccinic acid—to form a sandwich immunocomplex. The created immunosensor was characterized by a high sensitivity, good selectivity, and reproducibility.

### 4.3. One-Component Analysis

Most of the proposed immunosensors concerned the detection and/or determination of only one cancer biomarker. Such approaches were implemented based on the diagram shown in Figure 3 (in the portion marked with a red dotted line). In this context, both the substrate and the nanoparticles were functionalized with antibodies characteristic of a specific antigen. The measured signal was related only to the nanoparticles used, and the effect of nonspecific binding was considered in only a few cases. Two-thirds of the studies conducted concerned the analysis of one disease biomarker (details of the analyzed markers and the structures are included in Table 1).

### 4.4. Multiplexed Analysis

In the course of research on sandwich SERS immunosensors, appropriate modifications were made that led to the possibility of the multiplex analysis. Such works consisted of developing an appropriate procedure for the preparation of SERS substrates, which had to be immobilized with individual antibodies in the proper order. The most important advantage of such prepared sensors is the increase in the reliability of detecting a given cancer disease. For example, the detection of three biomarkers characteristic of a given disease simultaneously increases the probability of the occurrence of that disease three times. The simultaneous analysis of several antigens also shortens the analysis time and provides more information than the traditional approach. Moreover, it is possible to test more samples at the same time.

Among all publications concerning the detection/determination of cancer antigens, a multicomponent analysis was present only in one-third of all reports. This may be associated with the increased degree of complexity in the preparation of the appropriate immunosensors. Most of them represented the analysis of two biomarkers—15 literature reports [29,31,32,38,42,45,46,51,53,65,68,73,79,82,92]. Three cancer antigens were analyzed simultaneously in eight articles [28,40,48,69,74,80,83,91]. More biomarkers were taken into account only twice [81,95]. As mentioned previously, the most frequently analyzed antigens were AFP, CEA, and PSA. More information is included in Table 1.

## 5. Application of Flow Techniques

In recent years, there has been a huge increase in the application of microfluidics, lab-on-a-chip (LOC) systems, and devices operating on the lateral flow assay (LFA) principle in the context of various chemical analyses [98,99,100]. It has been shown that the use of technology that enables fluid manipulation on a submillimeter scale can be successfully applied in medical diagnostics, as well as biological/biomedical research. Due to some of the properties of microfluidic technology—fast sample processing, precise fluid control, limitation of the consumption of all required solutions, integration of many chemical processes on a single chip, and full automation, as well as maximization of information collected from valuable samples—these systems have become a very attractive approach that allows the replacement of traditional tests. Flow technologies are often adapted to enhance the capacity of researchers in biology and medical research.

The application of the microfluidic technique in the implementation of the enzyme immunoassay ELISA based on blood samples in the context of general health care has been presented [101]. These methods found clinical application because they use very small amounts of biological fluids to process the samples, and the analyses are performed quickly and easily. They have been widely used in cell analysis, isolation/detection of circulating neoplastic cells in the blood/urine of patients, and detection of biomarkers of many diseases or interactions between proteins [102,103].

The combination of the microfluidic technique with SERS detection allows for obtaining accurate and repeatable results with a very high sensitivity of the determinations. Such a combination was used, among others, to detect neoplastic cells and pathogens [103]; to monitor chemical reactions; in pharmaceutical, environmental, and biological analyses; and in medical diagnostics [104,105,106,107,108].

The use of microfluidic, lab-on-a-chip, and lateral flow systems in combination with sandwich-type SERS immunosensors in the detection of cancer biomarkers is illustrated below. Figure 5, Figure 6 and Figure 7 show examples of systems used to produce sandwich-type immunosensors in flow-through mode.

### 5.1. Microfluidics Technique

Several reports in the literature indicated the use of the microfluidic technique coupled with SERS spectroscopy based on the principle of sandwich immunosensors for the detection of substances indicative of the possible occurrence of cancer diseases. These platforms can be distinguished into two groups: continuous flow and flow-through; and segmented or droplet-based platforms. Among these reports, the system used to study the cancer biomarker carcinoembryonic acid, schematically presented in Figure 5, can be mentioned [87].

The main idea of this system is based on the integration of magnetic focusing and SERS detection in a microfluidic platform. A bifunctional nanocomposite probe consisting of a magnetic nickel–iron core and a gold shell (NiFe@Au) was used to detect the CEA biomarker. Bioconjugated probes with a capture antibody, as well as bioconjugated Au nanoparticles (30 nm) with a Raman reporter (4-mercaptobenzoic acid) and a detection antibody, together with biomarkers, were mixed and introduced into the microfluidic system. The resulting sandwich conjugates (NiFe@Au–CEA–Au), which provided enrichment of ‘hot spots’, were accumulated and dried in a microfluidic channel using a magnetic field before the SERS measurements. A Raman spectrometer with a 780 nm laser combined through the 5 mm channel of a microfluidic flow system was used to register the analytical signals. The determined CAE detection limit was set at about 0.1 pM, while the linear range was 0–1 ng mL^−1^. However, it should be considered that the determinations’ sensitivities depended on the size of the gold nanoparticles and the magnetic focusing time [87].

Lee et al. [26] developed a gold-array-embedded gradient chip. Sandwich immunocomplexes were created on the surface of 5 × 5 round on 30 µm gold-patterned microarray wells placed in the gradient channel of the microfluidic system under continuous conditions. The system was fully automatic and programmable, and provided a stable and reproducible SERS signal for the model cancer biomarker—α-fetoprotein. Hollow gold nanospheres were used as a SERS tag. The formation process of the immunocomplexes was automatically generated and controlled in the appropriately designed microfluidic channels at sizes of 100 mm (height) and 150 mm (width). One of the most important advantages was the possibility of automatic gradual dilution to generate a wide range of concentrations. In this way, the entire procedure, including incubation, washing, and detection, took less than 60 min.

The device consisted of three layers: a PDMS panel enabling a uniform distribution of HGN conjugates with antibodies (top layer), a PDMA panel for gradual dilution and generation of various AFP concentrations (middle layer), and a glass substrate with six gold array sets as a place to create the sandwich immunocomplexes (bottom layer). Briefly, the procedure for producing the immunocomplex was based on gold-patterned microarray immobilization with anti-AFP antibodies after previous immobilization with carboxylic acid and their activation with an EDC/NHS mixture. Next, after blocking unbound sites with BSA solution, the array was incubated with the AFP antigen (various concentrations were generated in the microfluidic chip). In the next step, PBS solution was used to remove the unbound antigens. Finally, HGN functional nanoprobes with polyclonal detection antibodies were introduced into the microfluidic system to produce the desired immunocomplexes. ‘Hot spots’ were generated not only between the edges of nanoparticles and planes of substrates, but also at the pinholes in the hollow gold nanospheres. A quantitative analysis was enabled by the application of a stop-flow mode to allow sufficient time for the production of the immunocomplex (the SERS signal was eight times higher than during the continuous-flow mode). The developed method enabled a good linear response in the range of 0 ng mL^−1^ to 10 ng mL^−1^ and an LOD for the rabbit AFP antigen equal to 0–1 ng mL^−1^, with a highly sensitive and reproducible SERS signal [26].

An interesting approach based on a fast, simple, and specific immunoassay with nanoscaled surface-shear forces was also carried out using the microfluidic technique in the detection of human epidermal growth factor receptor 2, which is an important biomarker in breast cancer [30]. The fluorophore-integrated gold/silver nanoshells were applied as SERS nanotags, while malachite green isothiocyanate constituted the Raman reporter. Enhancement of the capture kinetics was provided by the use of alternating current electrohydrodynamic (AC-EHD)-induced nanoscaled surface-shear forces. For the realization of the SERS immunoassay, a multiplexed device enabling fluid control and creation of the capture domain was constructed. This system consisted of an array of asymmetric electrode pairs and three microfluidic channels (each channel had 50 asymmetric planar electrode pairs) combined with gold connecting pads, which created the cathode and anode. The AC-EHD field was responsible for the movement of the sample and the SERS tag in the microfluidic device. The constructed system enabled a fast analysis (40 min), a high sensitivity (LOD of 10 fg mL^−1^), and very specific detection of HER2 at concentrations of 1 fg mL^−1^ to 1 ng mL^−1^ in biological samples of human serum [30].

Li and coworkers [92] published studies regarding 3D ordered silver nanoshell silica photonic crystal beads (Ag–SPCBs) as an encoded SERS substrate for multiplexed analysis of the carcinoembryonic antigen and α-fetoprotein biomarkers as a sandwich format. A microfluidic system was used to synthesize monodispersed size-controlled SPCBs, which then were embedded with silver nanoparticles to generate ‘hot spot’ sites for SERS amplification. Antibodies against both markers were attached to the prepared substrates, and in the next step, cancer was included with one Raman label. In the end, in the last incubation stage, SERS-signal-amplified probes consisting of silver nanoparticles with the antibody and Raman reporter (4-MBA) were added to form a sandwich complex with good reproducibility (CV < 10%). The linear ranges were 0.01 pg mL^−1^ to 1000 ng mL^−1^ and 0.1 pg mL^−1^ to 1000 ng mL^−1^ for CEA and AFP, respectively. The achieved LOD value for CEA was 6.6 × 10^−6^ ng mL^−1^, and for AFP was equal to 7.2 × 10^−5^ ng mL^−1^. The practical utility of diagnostic tests was proved in the clinical human serum samples analysis by comparing the results with the reference method—an electrochemiluminescent immunoassay [92].

In the sandwich version, a pump-free microfluidic system chip was also proposed and constructed for the rapid and sensitive immunodetection of prostate-specific antigen biomarkers [78]. PSA detection antibody-conjugated gold SERS tags (malachite green isothiocyanate used as Raman reporter), PSA capture antibody-conjugated magnetic beads, and a cancer antigen were the main elements that, after being introduced into the microfluidic system, allowed for the formation of immunocomplexes. A permanent magnet integrated into the PDMS system was responsible for delivering the immunocomplex to the measuring chamber. This portable pump-free system enabled a fast analysis (5 min) of human serum (sample volume of 80 μL) without long incubation steps, a wide linear range (0.01 ng mL^−1^ to 100 ng mL^−1^), and a good detection limit (<0.01 ng mL^−1^). Both rapid prototyping and UV photolithography methods were used to fabricate the chip. The chip size was 40 mm by 21 mm, and it consisted of a part for sample mixing and immunoreaction (3 mm round chamber and winding shape structure), a compartment dedicated to storage and detection (1.6 mm length rectangular chamber), and a capillary pump (a comblike structure channel and a 4.0 mm round mini-magnet) [78].

Type 1 cytokine-like interleukins (IL-6, IL-8, and IL-18) present in blood plasma may indicate the presence of various diseases, including cancer. Kamińska et al. [28] proposed a microfluidic system for fast, sensitive, and multiplexed monitoring of these interleukins based on a combination of sandwich-type sensors with SERS detection. Bimetallic hybrid Ag–Au platforms were processed to receive SERS-active substrates immobilized with three different antibodies. Appropriate biomarkers (interleukins) constituted the second layer of the emerging immunocomplex. The detection nanoparticles were coated with three different Raman reporters (basic fuchsin, 5,5′-dithio-bis(2-nitro-benzoic acid), and p-mercaptobenzoic acid) and with antibodies to specific antigens/interleukins (third layer). The constructed microfluidic system (channels size: 200 or 400 µm width and 350 µm depth) enabled the carrying out of the procedure in two variants: a parallel method, with three chambers in the fluidic system, each leading to different interleukin (LOD values < 6.5 pg mL^−1^); and in a simultaneous multiplexed approach, in which one platform was immobilized with three antibodies that detected each considered interleukin (LOD values < 7.5 pg mL^−1^). The method’s effectiveness was confirmed by a quantitative human blood plasma analysis that was compared to the results obtained with an ELISA assay [28].

Unlike continuous flow microfluidics, the droplet version of microfluidics relies on the generation of discrete droplets (from femtoliters to microliters) and their manipulation inside the microflow system. The advantages of this approach include the possibility of miniaturized reactions in separate droplets, rapid mixing, fully controlled time of reactions, and highly uniform synthesis of micro- and nanostructures. The combination of droplet microfluidic systems with SERS techniques for cancer biomarker detection in sandwich immunoassays enabled rapid and reproducible analyses [77,82].

The first example of the application of a droplet microfluidic device concerned the development of a novel technique: a wash-free magnetic immunoassay applied in rapid and sensitive prostate-specific antigen detection and determination in serum [77]. The most important element of this research was the use of a rectangular magnetic bar inside the microflow system. This solution made it possible to separate the droplet into two unbound and bound magnetic SERS-tag parts. Separated magnetic immunocomplexes with structure capture antibody-conjugated magnetic bead-antigen-detection antibody-conjugated gold nanoparticles were subjected to SERS measurements. The more PSA antigens present in a single droplet, the more immunocomplexes were formed, which was the basis for the quantitative analysis. The assay procedure was fully automatic with measurement of analytical signals at 174 droplets min^−1^, a concentration range of 0.05 g mL^−1^ to 200 ng mL^−1^, and an estimated LOD below 0.1 ng mL^−1^.

Another example of a microfluidic droplet system with magnetic-field-amplified SERS involved research on rapid, sensitive (LOD = 1.0 fg mL^−1^ in one droplet, concentration range of 1.0 fg mL^−1^ to 10 pg mL^−1^), and simultaneous detection of two cytokines secreted by a single cell: vascular endothelial growth factor (VEGF) and interleukin-8 (potent angiogenic factors) [82]. Cross-typed microfluidics allowed the formation of water-in-oil droplets with individual cells and all types of immune particles: antibody-conjugated Ag nanoparticles and magnetic beads for both cytokines (four types of functionalized immune parts). Sandwich immunocomplexes were generated through antigen–antibody recognition, and were collected in a channel array for signal registration. Raman reporters (4-aminobiphenyl for VEGF and acetamide for IL-8) attached to magnetic beads resulted in an enhanced SERS signal after conjugation with silver nanoparticles. The effect of spontaneous collection induced by the magnetic field enabled another signal enhancement (75 times).

### 5.2. Lab-on-a-Chip (LOC)

Lee et al. [42] presented a highly sensitive and reproducible immunoassay for the model cancer biomarkers α-fetoprotein and angiogenin with the use of a smooth gold-patterned microarray chip consisting of 1 mm diameter 4 × 4 wells. Hollow gold nanospheres (HGNs) labelled with a Raman reporter (malachite green isothiocyanate) were created and immobilized with antibodies. Sandwich-immune complexes were created between HGN structures and gold-patterned wells. This approach enabled reproducible signals during SERS mapping in a wide range of concentrations (0 g mL^−1^–10^−4^ g mL^−1^), and had a dynamic range of 10^−4^ g mL^−1^–10^−12^ g mL^−1^ for SERS imaging. ‘Hot spot’-enhanced SERS signals arose from two sources: the edges between the nanospheres and the array chip substrate. The limits of detection for ANG and AFP were 0.1 pg mL^−1^ and 1.0 pg mL^−1^, respectively.

Another example involved the use of a specially designed chip for the parallel multiplexed analysis of cancer antigens: CA 15-3, CA 27-29, and the cancer embryonic antigen in serum, with improved sensitivity [69]. The diagram of the chip is shown schematically in Figure 6; it was designed in such a way as to enable the detection and determination of several cancer markers, which increased the certainty regarding the presence of a given disease entity. Both the chip panel and SERS tags (gold nanostars with 4-nitrothiophenol coated with a thin silica layer) were functionalized with antibodies against selected markers (detection antibody). A SERS chip with well-defined wells placed on a quartz substrate was functionalized with the antibody after attachment and activation of the carboxylic group (capture antibody); 2 µL standard solutions or serum samples spiked with different cancer markers concentrations were added to each well. Immuno-sandwich complexes were formed on the created chips due to specific interactions between the antigen and antibody of selected markers. The SERS platform consisted of a 3 × 3 array, with rows for three selected biomarkers and columns for triple measurements of the SERS signal (imaging) [69].

Wan and coworkers [25] presented a sensor with a sandwich configuration for prostate-specific antigen (PSA) analysis in human serum. The gold chip and SERS detection tags were used to form such an immunocomplex. Gold nanoparticles (~40 nm) functionalized with a Raman reporter (1,4-benzenedithiol) were covered with a thin layer of a gold-detection element that was ready to use. However, a fluidic channel was used for the realization of some steps of the sensor-formation procedure on the gold chip. The measurement was conducted using flow-injection surface plasmonic resonance (FI-SPR). The samples were introduced to the sample loop (200 mL) using a six-port valve and a syringe pump to a flow cell (1 mL). The LOD was equal to 2.0 pg mL^−1^, while the working concentrations ranged from 10 pg mL^−1^ to 10 ng mL^−1^.

### 5.3. Lateral Flow Assay (LFA)

A lateral flow assay (LFA) is a method used for low-cost, simple, portable, and rapid testing of the important substances in biomedical, food, and environmental analyses. In this type of research, a liquid sample or extract containing the analyte (biomarker) moves through the different zones of the polymer strips (pad). The relevant molecules on such a strip interact with the biomarkers, signaling the presence of a given analyte. In one variation of this method, the antibodies are used as recognition elements. This technique’s unique advantages (low cost and ease of production) predispose it to detect disease markers and infectious agents [109]. Increases in sensitivity and reproducibility and improving multicomponent analysis are continuing challenges. In this context, the use of SERS spectroscopy is highly desirable in the early stages of cancer.

A SERS-based lateral flow immunoassay for squamous cell carcinoma antigen and cancer antigen 125 in the cervical cancer serum was proposed recently [68]. Figure 7 shows a schematic strip with a test and control line used in this research. After functionalization with an antibody and Raman reporter (4-nitrothiophenol and 5,5′-dithiobis(2-nitrobenzoic acid)), nano-silver polydopamine nanospheres (70 nm) were applied as detection tags. On the test line, the aggregation of nanospheres took place by the formation of sandwich-configuration immunocomplexes as a result of binding to the antigen attached to the antibodies located on the T line (the color became darker). The test only worked correctly when the control line became colored. A high sensitivity was proved for LOD values, which were estimated to be as low as 8.093 pg mL^−1^ and 7.370 pg mL^−1^ in human serum for SCCA and CA125, respectively. The LFA assay enabled a high specificity and reproducibility, with a full time for the complete procedure of 20 min. The clinical usefulness was confirmed using an analysis of the serum samples taken from patients by comparing the results with the ELISA test [68].

## 6. Conclusions and Future Perspectives

Surface-enhanced Raman spectroscopy exhibits many advantages compared to traditional techniques, such as ultrahigh sensitivity, very good levels of multiplexing, robustness, and the ability to perform detection and determination of analytes in biological samples with a complex matrix. In turn, sandwich-type immunoassays are characterized by their high sensitivity, simplicity, cost-effectiveness, and speed of analysis. Therefore, the combination of SERS techniques with immunosensing represents a very prospective approach, with particular potential application as a point-of-care (PoC) testing tool, allowing the detection of even a single molecule.

The present review concentrated on recent advancements in SERS immunosensing techniques, such as sandwich immunosensors, to detect and determine different types of cancer biomarkers. We have presented the principle of the sandwich SERS immunosensor and its construction, with a description of the SERS substrates and nanoparticles used. The applications in the detection of cancer biomarkers have been described in detail, along with the analytical characteristics of single-component and multiplexed analyses. The methods of signal detection in terms of the use of a single immunoprobe, as well as dual-tag systems, were also discussed. Additionally, the methods of implementing this type of immunoassays in flow systems were characterized.

The SERS sandwich-type immunoassay is an excellent tool for detecting cancer biomarkers in biological samples with a high sensitivity and low limits of detection. Immunosensors can be used with several analytes of interest. However, the maximum number of markers detected so far is five. The use of three biomarkers increases the complexity of the preparation of these sensors. Both substrates and nanoparticles affect the possibility of amplification of the SERS signal. They are characterized by different architectures and constructions (considering metallic, nonmetallic, or mixed natures, as well as magnetic or nonmagnetic). These systems are most often implemented using the version in which the Raman reporter is combined only on the detection element, which is associated with some disadvantages. Only the dual-tag approach (with the reporter on both the substrate and the nanoparticle) allows for excluding false-positive signals resulting from nonspecific bonds or aggregation of nanoparticles on the sensor surface.

The use of flow-through techniques (microfluidics, lab-on-a-chip, lateral flow assay) to implement these sensors has many advantages, including fast sample processing, precise fluid control, less consumption of all required solutions, integration of various chemical processes on a single chip, automation, and an increase in information from valuable samples.

Researchers will continue to search for new materials/architectures as candidates for substrates and nanoparticles in the coming years. Such materials should ensure high amplification of the recorded SERS signal and its stability. An important aspect is also the possibility of regenerating the sensor surface, on which scientists for many research groups worldwide are already working. Particles prepared using core–shell technology seem to be particularly promising, while molecularly imprinted polymers can be considered as candidates for substrates; an interesting solution may be the discovery of new shapes of nanoparticles composed not only of gold and/or silver, but also alloyed with other metals. In the coming years, research will also be carried out on faster and more efficient methods of synthesizing individual elements included in sandwich-type SERS immunosensors and improving the processes of the formation of these sensors. It will be necessary to optimize the size of the sensors to be able to perform SERS measurements simply and quickly.

Due to the credibility of the information obtained, the possibility of detecting more biomarkers is the most desirable aspect. Sensors enabling multiplex analysis will be developed increasingly often. However, we anticipate that such sensors will perform best when detecting three to five antigens. Focusing on more cancer markers will be very problematic, beginning with the preparation of the immunosensor itself, for which the complexity of the preparation procedure increases significantly. However, the problem will also be related to the analysis and processing of signals, and correlating these signals well with the concentration of the detected substances. In the authors’ opinion, it may be helpful to develop chemometric analysis algorithms, which will facilitate the interpretation of the obtained SERS signals. In the future, universal tools should be generated from the point of view of chemometric analysis, as well as with the use of artificial intelligence, more or less developed for this purpose. It will be necessary to focus in new works on the use of a double-tag system to fully eliminate false-positive signals. The chemometric analysis will be able to analyze whether there is nonspecific binding between the antigen and the antibody in the analytical system, whether there is an aggregation of nanoparticles on the surface, or whether there are other types of interference effects. By applying the analysis from a methodological point of view, it will be possible to increase the credibility of the obtained results, which will improve the diagnostic process of neoplastic diseases.

In future research, microfluidic systems and portable lateral flow assays (LFAs) will be used increasingly often. Their use will reduce the consumption of reagents and samples, and fully automate the production of immunocomplexes, which will positively affect the repeatability of the sensor formation, and thus the repeatability of the analysis. Due to the possibility of easy portability, low cost, and simple LFA construction, the systems will prove very useful information in preliminary investigations using portable Raman spectrometers.

Indeed, the research described in this paper and future innovations will enable quick and reliable detection of neoplastic diseases. However, diagnostic tests based on SERS sandwich-type immunosensors may be a better alternative to the currently used ELISA tests.

## Figures and Tables

**Figure 1 ijms-23-04740-f001:**
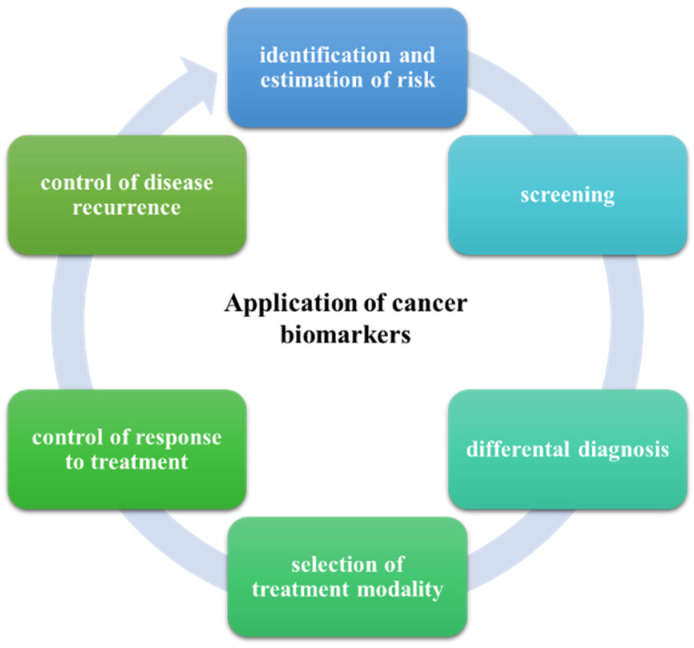
Exemplary applications of cancer biomarkers for clinical and medical purposes (based on [13,14]).

**Figure 2 ijms-23-04740-f002:**
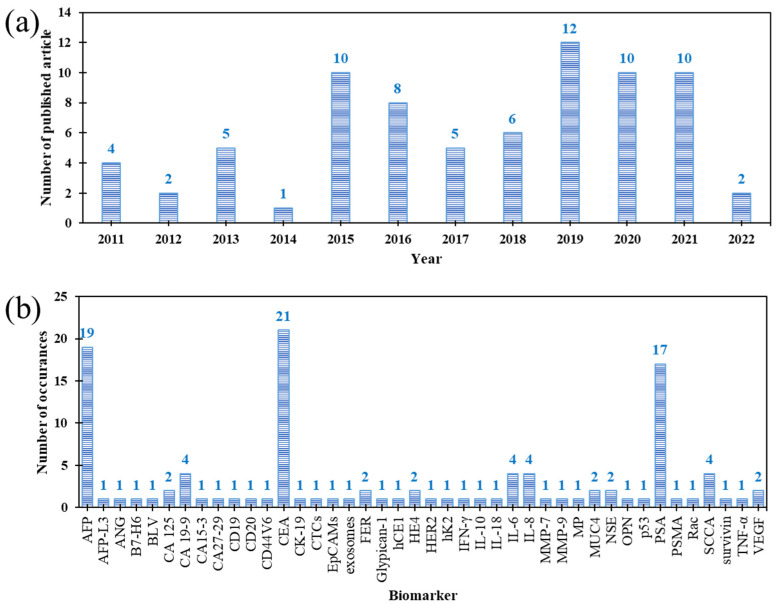
(**a**) Number of published articles during the period of 2011 to 2022 (February) based on Scopus, ScienceDirect, PubMed, Web of Science, and Google Scholar. (**b**) Number of biomarker occurrences in the published articles, where: AFP—α-fetoprotein; AFP-L3—lectin-reactive α-fetoprotein; ANG—angiogenin; B7-H6—B7-H6 protein; BLV—bovine leukemia virus antigen gp51; CA 125—carbohydrate antigen 125; CA 19-9—carbohydrate antigen 19-9; CA15-3—cancer antigen 15-3; CA27-29—cancer antigen 27-29; CD19—specific surface marker CD19; CD20—specific surface marker CD20; CD44V6—CD44 variant isoform 6; CEA—carcinoembryonic antigen; CK-19—cytokeratin-19; CTCs—circulating tumor cells; EpCAMs—epithelial cell adhesion molecules; exosomes—tumor-derived exosomes; FER- ferritin; hCE1—human carboxylesterase 1; HE4—human epididymis protein 4; HER2—human epidermal growth factor receptor 2; hK2—human kallikrein 2; IFN-γ—interferon gamma; IL-10—interleukin-10; IL-18—interleukin-18; IL-6—interleukin-6; IL-8—interleukin-8; MMP-7—matrix metalloproteinase-7; MMP-9—matrix metalloproteinase-9; MP—metanephrine; MUC4—mucin protein MUC4; NSE—neuron-specific enolase; OPN—osteopontin; p53—protein p53; PSA—prostate-specific antigen; PSMA—prostate-specific membrane antigen; Rac—adrenal stimulant ractopamine; SCCA—squamous cell carcinoma antigen; TNF-α—tumor necrosis factor α; VEGF—vascular endothelial growth factor.

**Figure 3 ijms-23-04740-f003:**
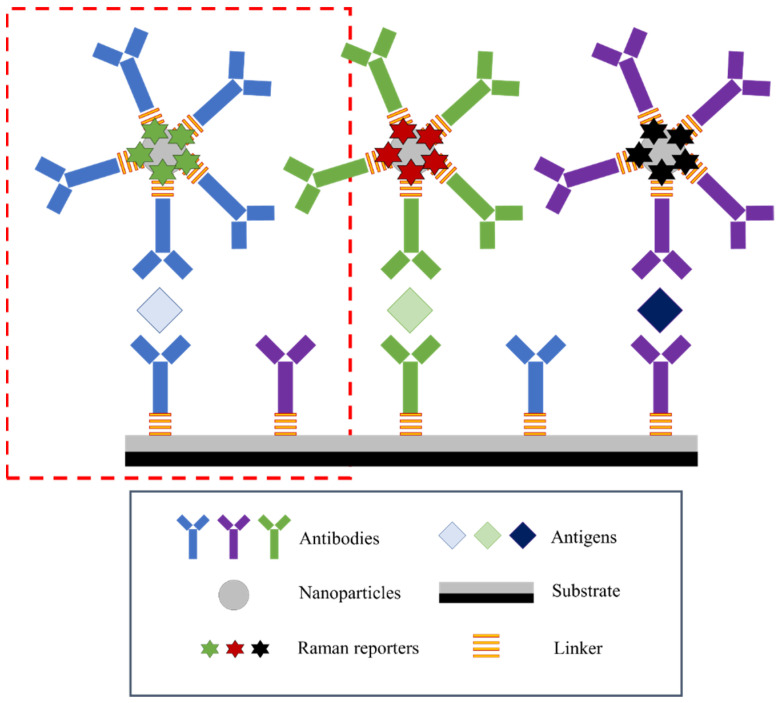
Schematic construction of the sandwich SERS immunosensor in the multiplexed version. The most frequently used approach enabling single-antigen analysis is shown in the red box.

**Figure 4 ijms-23-04740-f004:**
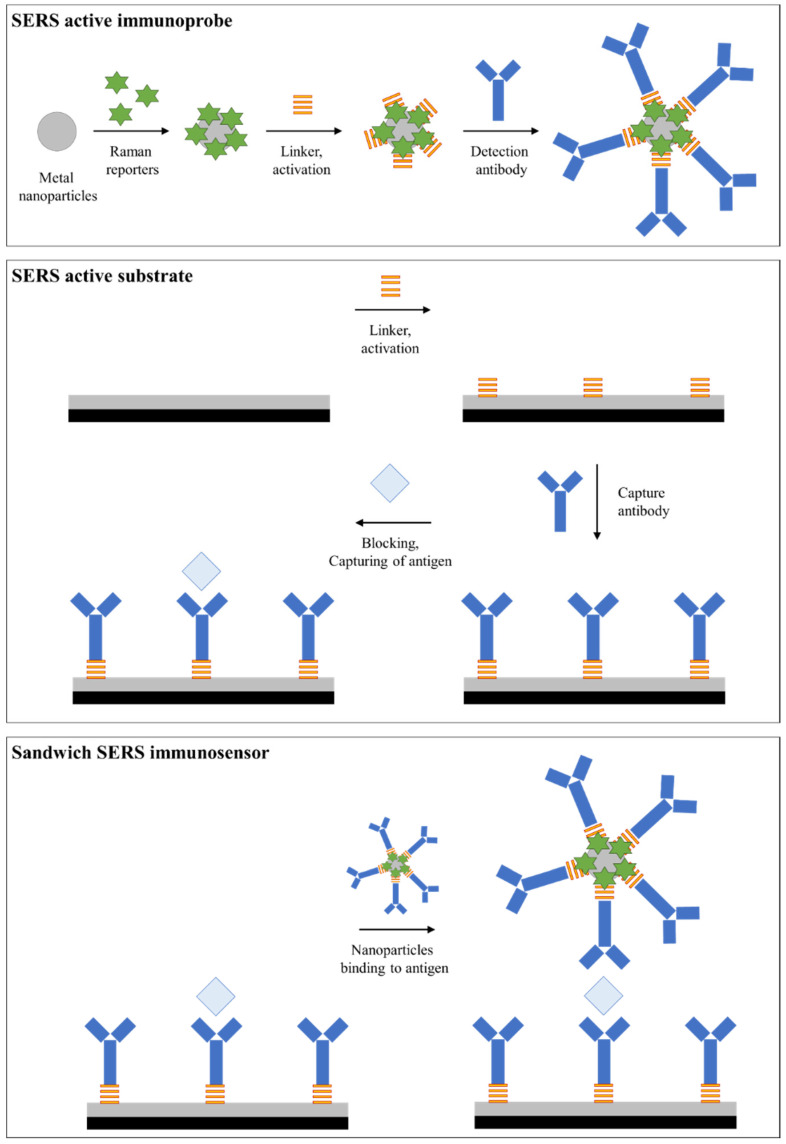
Diagram showing the steps of developing a sandwich-type SERS immunosensor: immunoprobe and substrate preparation, and the immunosensor functioning.

**Figure 5 ijms-23-04740-f005:**
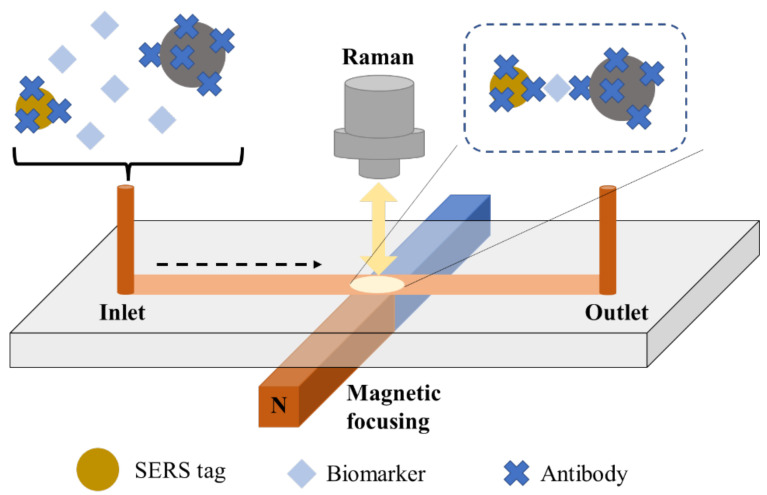
Schematic representation of microfluidic system combined with SERS for biomarker detection based on antigen–antibody interactions (based on [87]).

**Figure 6 ijms-23-04740-f006:**
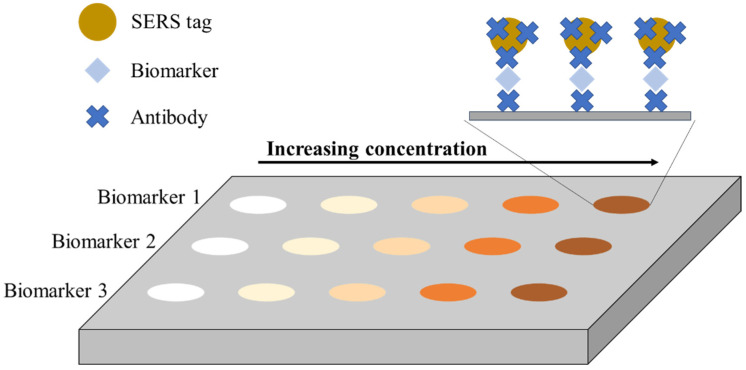
Schematic representation of lab-on-a-chip platform combined with SERS for biomarker detection based on antigen–antibody interactions (based on [69]).

**Figure 7 ijms-23-04740-f007:**
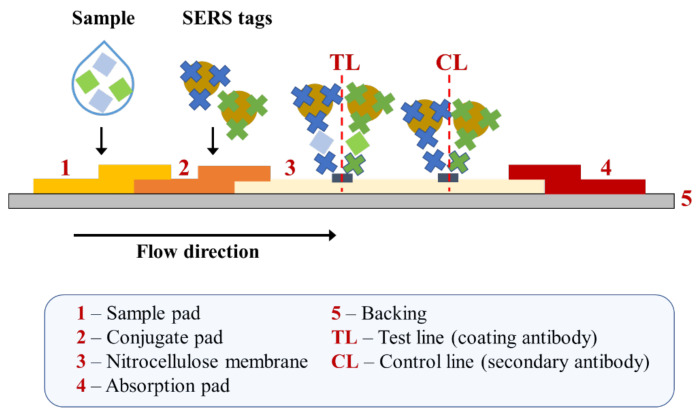
Schematic representation of lateral flow assay combined with SERS for biomarker detection based on antigen–antibody interactions (based on [68]).

**Table 1 ijms-23-04740-t001:** Exemplary studies based on different SERS sandwich immunosensor designs for cancer detection resulting in different detection approaches.

Ref.	Biomarker (Multiplexing Ability)	Single Tag (St)/Dual Tag (Dt) (Raman Reporters)	Nanoparticles (NPs)	Platform	Dynamic Range	LOD
[43]	PSA	**St** (R6G)	Colloidal AuNPs	AuNP layer of the patterned substrate	-	-
[27]	PSA	**St** (MBA)	Nano-Ag immune probes	Nano-Ag/Au immune substrate	-	1.8 fg mL^−1^
[36]	PSA	**St** (MBA)	Silica-coated Ag nanorods (NRs)	Quartz slide with Ag nanorods (NRs)	0.3 fg mL^−1^–3 µg mL^−1^	0.3 fg mL^−1^
[77]	PSA	**St** (MGITC)	AuNPs	Magnetic beads (MB)	50 pg mL^−1^–200 ng mL^−1^	0.1 ng mL^−1^
[25]	PSA	**St** (1,4-BDT)	Au core–Au shell NPs	Au plate	10 pg mL^−1^–10 ng mL^−1^	2.0 pg mL^−1^
[39]	PSA	**St** (TB)	ZnO and CoFe_2_O_4_ nanocomplexes with Au	Si@Ag substrate	1 pg mL^−1^–10 ng mL^−1^	0.65 pg mL^−1^
[64]	PSA	**St** (GO)	AgNPs deposited on graphene oxide (GO–AgNP)	Polystyrene 96-well plate substrate	0.5 pg mL^−1^–500 pg mL^−1^	0.23 pg mL^−1^
[78]	PSA	**St** (MGITC)	AuNPs	MB	0.01 ng mL^−1^–100 ng mL^−1^	0.01 ng mL^−1^
[58]	PSA	**St** (MBA)	AuNPs	Silver-NP-decorated electrospun polymeric fibers	1–10 pg mL^−1^	1 pg mL^−1^
[50]	PSA	**St** (MBA)	Au seeds on Fe_3_O_4_@TiO_2_ core–shell NPs	Ag-coated sandpaper	10^−4^–10^−12^ g mL^−1^	0.014 mM
[63]	PSA	**St** (DNTB)	AuNPs	Magnetic molecularly imprinted polymers (MMIPs)	0.5 pg mL^−1^–1.0 μg mL^−1^	0.9 pg mL^−1^
[26]	AFP	**St** (MGITC)	Hollow gold nanospheres (HGNs)	Gold array	0–10 ng mL^−1^	0–1 ng mL^−1^
[34]	AFP	**St** (MBA)	AuNPs	Glass slide modified with AuNPs	1–100 ng mL^−1^	100 pg/mL
[60]	AFP	**St** (MPBA)	AgNPs	Molecularly imprinted polymer (MIP) array	1 ng mL^−1^–10 µg mL^−1^	-
[52]	AFP	**St** (MBA)	Core-shell SiO_2_@Ag	Ag-decorated NiCo_2_O_4_ nanorods	2.1 fg mL^−1^–2.1 ng mL^−1^	2.1 fg mL^−1^
[41]	AFP	**St** (MBA)	Ag-covered polystyrene sphere (PS@Ag)	Deposited Si pyramid array (Si@Ag) substrate	2 fg mL^−1^–200 ng mL^−1^	1.75 fg mL^−1^
[66]	AFP	**St** (MBA)	Silica-coated gold/silver core–shell nanostars (AuNS@Ag@SiO_2_)	Nitrocellulose (NC) membrane	3 pg mL^−1^–3 µg mL^−1^	0.72 pg mL^−1^
[67]	AFP	**St** (MBA)	Nanosphere with a silver coating core (Au@Ag), ultrathin continuous silica (SiO_2_) shell, and high coverage of gold nanosphere (AuNP) satellites	Nitrocellulose (NC) membrane	1 fg mL^−1^–1 ng mL^−1^	0.3 fg mL^−1^
[89]	AFP	**St** (MBA)	Gold-coated silver nanoparticles (Ag@AuNPs)	Boric-acid-functionalized magnetic silica particles	1.0 ng mL^−1^–1.0 mg mL^−1^	1.0 ng mL^−1^
[71]	AFP	**St** (R6G)	Silver-coated gold nanocubes	Molybdenum disulfide (MoS_2_)	1 pg mL^−1^–10 ng mL^−1^	0.03 pg mL^−1^
[75]	CEA	**St** (MBA)	HGNs	Magnetic microspheres	0 ng mL^−1^–100 ng mL^−1^	10 pg mL^−1^
[87]	CEA	**St** (MBA)	AuNPs	Au-coated NiFe magnetic nanoparticles (NiFe@Au)	0 ng mL^−1^–1 ng mL^−1^	0.1 pM
[94]	CEA	**St** (MBA)	AuNPs	γ-Fe_2_O_3_@AuNPs	1 ng mL^−1^–50 ng mL^−1^	0.1 ng mL^−1^
[33]	CEA	**St** (Nile blue)	AuNPs with polydopamine resin (PDR)	Chitosan-stabilized AuNPs on a glassy carbon electrode (GCE)	1 pg mL^−1^-100 ng mL^−1^	0.68 pg mL^−1^
[61]	CEA	**St** (MPBA)	AuNPs	Boronate affinity molecularly imprinted polymer (MIP) array	0.1 ng mL^−1^–1 mg mL^−1^	0.1 ng mL^−1^
[62]	CEA	**St** (ATP)	Gold nanostars (AuNSs)	Molecularly-imprinted polymer (MIP) film	0–1000 ng mL^−1^	1.0 ng mL^−1^
[57]	CEA	**St** (MBA)	MoS_2_ nanoflowers@AuNPs	Fe_3_O_4_@AuNP-functionalized delaminated Ti_3_C_2_T_x_ MXene-magnetic supporting substrate	0.0001–100.0 ng mL^−1^	0.033 pg mL^−1^
[8]	CEA	**St** (ATP)	AuNSs	Screen-printed electrode (Au-SPE)	0.025–250 ng mL^−1^	0.025 ng mL^−1^
[54]	SCCA	**St** (MBA)	Gold nanocages (GNCs)	Gold-nanoparticle-coated polydopamine resin microspheres (PDR@GNPs)	1 × 10^−5^ M–1 × 10^−10^ M	7.16 pg mL^−1^
[47]	MUC4	**St** (NBT)	AuNPs	Template-stripped gold (TSG)	0 μg mL^−1^–1 μg mL^−1^	33 ng mL^−1^
[35]	MUC4	**St** (NBT)	AuNPs	Glass chip with a gold layer	0.1 μg mL^−1^–20 μg mL^−1^	0.1 μg mL^−1^
[90]	IL-6	**St** (covalent conjugation of DTNB to a short MEG–OH and a longer TEG–CO_2_H group)	AuNPs	Gold/silver nanoshells (Au/AgNSs)	1 pg mL^−1^–1 µg mL^−1^	1 pg mL^−1^
[9]	IL-6	**Dt** (MBA, NTP)	Ag and AuNPs	Au and Ag hexagonal nanoarray	0 pg mL^−1^–1000 pg mL^−1^	25.2 pg mL^−1^
[93]	IL-8	**St** (MBA)	GNCs	Highly branched gold nanoparticle (HGNP) substrates	10 pg mL^−1^–1 µg mL^−1^	6.04 pg mL^−1^
[44]	VEGF	**St** (MGITC)	AuNSs	Gold triangle nanoarray	0.1 pg mL^−1^–10 ng mL^−1^	1.158 ng mL^−1^
[85]	BLV	**St** (DTNB)	Au rods	Magnetic gold NPs (MNP-Au)	0 mg mL^−1^–0.06 mg mL^−1^	0.95 µg mL^−1^
[84]	HE4	**St** (MBA)	AuNPs	Magnetic core–shell Fe_3_O_4_@AuNPs	1 pg mL^−1^–10 ng mL^−1^	100 fg mL^−1^
[24]	HE4	**St** (MGITC)	AuNPs	Gold (Au) nanoplate (NPl)	0 M–10^−9^ M	10^−17^ M
[30]	HER2	**St** (MGITC)	Gold/silver nanoshells	Gold electrode surface	1 fg mL^−1^–100 pg mL^−1^	10 fg mL^−1^
[76]	tumor-derived exosomes	**St** (DTNB)	Gold core–silver shell nanorods (Au@AgNRs)	MB	4.88 × 10^6^–4.88 × 10^3^	1200 exosomes
[37]	metanephrine	**St** (pATP, CV)	AuNPs	Au films on microscope glass slides	10^−3^ M–10^−5^ M	10^−4^ M
[70]	p53	**St** (ATP)	AuNPs	Glass substrate	10^−10^ M–10^−17^ M	10^−15^ M
[55]	hCE1	**St** (MBA)	AgNPs	Raspberry-like morphology of Fe_3_O_4_@SiO_2_@AgNP magnetic nanocomposites	0.1 ng mL^−1^–1.0 mg mL^−1^	0.1 ng mL^−1^
[56]	B7-H6 biomarker	**St** (ATP)	Spiky AuNPs	Au thin film modified with a self-assembled monolayer of zwitterionic L-cysteine	10^−10^ M–10^−14^ M	10^−14^ M (10.8 fg mL^−1^)
[49]	FER	**St** (4MBA)	Gold (Au)-coated ‘stellate’ mesoporous SiO_2_@Au nanoprobe	Hydrophilic Ag-deposited sandpaper assembled with hydrophobic-treated filter paper (coffee-ring-free hydrophilic–hydrophobic substrate)	1 × 10^−5^ g mL^−1^–3 × 10^−13^ g mL^−1^	3.16 × 10^−14^ g mL^−1^
[88]	MMP-9	**St** (DTNB)	AgNPs	Fe_3_O_4_ microspheres (magnetic NPs)	0 ng mL^−1^–100 ng mL^−1^	1 pg mL^−1^
[59]	CA19–9	**St** (MBA)	Immunoprobe of anti-CA19-9/4-MBA	Au nanowires (NWs) onto Fe_3_O_4_@TiO_2_ matrix	1000 IU mL^−1^–0.001 IU mL^−1^	5.65 × 10^−4^ IU mL^−1^
[72]	CTCs	**St** (GO)	Gold–graphene hybrid nanotag (Au–rGO)/gold-reduced graphene oxide nanosystem)	Polycarbonate filter	1 cell mL^−1^–100 cell mL^−1^	1 cell mL^−1^
[42]	ANG, AFP (**2**)	**St** (MGITC)	HGNs	Gold-patterned microarray chip	0 g mL^−1^–10^−4^ g mL^−1^	0.1 pg mL^−1^ (ANG), 1.0 pg mL^−1^ (AFP)
[73]	CEA, AFP (**2**)	**St** (MGITC, XRITC)	HGNs	MB	-	-
[92]	CEA, AFP (**2**)	**St** (MBA)	AgNPs	3D ordered silver nanoshell silica photonic crystal beads (Ag-SPCB)	0.01 pg mL^−1^–1000 ng mL^−1^ (CEA), 0.1 pg mL^−1^–1000 ng mL^−1^ (AFP)	6.6 × 10^−6^ ng mL^−1^ (CEA), 7.2 × 10^−5^ ng mL^−1^ (AFP)
[31]	CEA, AFP (**2**)	**St** (MB, TMB)	AuNPs	Gold microelectrode array (GMA)	0.01 ng mL^−1^–20 ng mL^−1^ (CEA), 0.02 ng mL^−1^ 0–5 ng mL^−1^ (AFP)	0.3 pg mL^−1^ (CEA), 0.6 pg mL^−1^ (AFP)
[46]	CEA, AFP (**2**)	**St** (MBA, DTNB)	AuNSs	Ordered gold nanohoneycomb arrays	0.5 ng mL^−1^–100 ng mL^−1^	0.41 (CEA), 0.35 ng mL^−1^ (AFP)
[32]	CEA, CK-19 (**2**)	**St** (THI, NBA)	AuNP-coated acid-based resin (AAR) microspheres	Electrode-modified chitosan-stabilized AuNPs	0.05 ng mL^−1^–80 ng mL^−1^	0.01 ng mL^−1^ (CEA), 0.04 ng mL^−1^ (CK-19)
[45]	CA 19-9, MMP-7 (**2**)	**St** (DSNB)	AuNPs	Array of exposed gold ‘wells’	-	2.28 pg mL^−1^ (MMP-7), 34.5 pg mL^−1^ (CA 19-9)
[65]	PSA, Rac (**2**)	**St** (MBA)	Aggregated AgNPs	96-Well polystyrene plates	-	10^−9^ ng mL^−1^ (PSA), 10^−6^ (Rac) ng mL^−1^
[86]	CEA, NSE (**2**)	**St** (MBA, DTNB)	Flowerlike gold NPs	Gold-coated magnetic nanoparticles	10 pg mL^−1^–100 ng mL^−1^	1.48 pg mL^−1^ (CEA), 2.04 pg mL^−1^ (NSE)
[53]	PSA, AFP (**2**)	**St** (NTP, MBA)	AgNPs coated on SiO_2_ nanospheres (SiO_2_@Ag)	Gold-film hemisphere array (Au-FHA) immune substrate	10 fg mL^−1^–400 ng mL^−1^	3.38 (PSA), 4.87 (AFP) fg mL^−1^
[79]	CD19, CD20) (**2**)	**St** (MBA, (DNTB)	AgNPs	MB	5000 cells mL^−1^–5 cells mL^−1^	5 cells mL^−1^
[38]	AFP, AFP-L3 (**2**)	**Dt** (MBA, DSNB)	AuNPs with DSNB	Silicon chips coated with Ag (Si/Ag/MBA)	0.5 ng mL^−1^–1000 ng mL^−1^	0.5 ng mL^−1^
[82]	VEGF, IL-8 (**2**)	**St** (ABP, AAD)	AgNPs	MB	1.0 fg mL^−1^–1 ng mL^−1^	1.0 fg mL^−1^
[68]	SCCA, CA125 (**2**)	**St** (ATP, DTNB)	Nano-Ag polydopamine nanospheres (PDA@Ag-NPs)	Nitrocellulose (NC) membrane	10 pg mL^−1^–10 µg mL^−1^	7.156 pg mL^−1^ (SCCA), 7.182 pg mL^−1^ (CA125)
[51]	SCCA, OPN (**2**)	**Dt** (MBA, DTNB, DMSA)	Au–Ag nanoshuttles (Au–AgNSs)	Hydrophobic filter-paper-based Au nanoflowers (AuNFs)	10 pg mL^−1^–10 µg mL^−1^	8.628 pg/mL (SCCA), 4.388 pg/mL (OPN)
[29]	SCCA, survivin (**2**)	**St** (DTNB, ATP)	Au–Ag nanoshells (Au–AgNSs)	Au–Ag nanobox (Au-AgNB) array substrate	10 pg mL^−1^–10 µg mL^−1^	6 pg mL^−1^ (SCCA), 5 pg mL^−1^ (survivin)
[74]	CEA, AFP, CA 125 (**3**)	**St** (3-MeOBT, 2-MeOBT, 2-NT))	Nanotags with hybrid multilayered nanoshells prepared using layer-by-layer (LBL) assembly of small silver nanoparticles (AgNPs) at the surface of silica (SiO_2_) particles using poly(ethyleneimine) (PEI)	MB	0.1 ng mL^−1^–1 ng mL^−1^	0.1 pg mL^−1^
[69]	CA 15-3, CA 27-29, CEA (**3**)	**St** (NTP)	AuNSs	Quartz chip with punched wells	0.1 ng mL^−1^–500 ng mL^−1^	0.99 U mL^−1^ (CA 15-3), 0.13 U mL^−1^ (CA 27-29), 0.05 ng mL^−1^ (CEA)
[48]	PSA, AFP, CA19-9 (**3**)	**St** (MBA)	SiO_2_-coated Si nanoparticles	SiC@Ag substrate (Ag film sputtered on SiC sandpaper)	0–5 mg mL^−1^ (PSA, AFP), 0–3 mg mL^−1^ (CA19-9)	1.79 fg mL^−1^ (PSA), 0.46 fg mL^−1^ (AFP), 1.3 × 10^−3^ U mL^−1^ (CA19-9)
[40]	PSA, PSMA, hK2 (**3**)	**St** (MBA)	AgNPs	SiC@Ag@Ag-NPs substrates	10^−5^–10^1^ ng mL^−1^	0.46 fg mL^−1^ (PSA), 1.05 fg mL^−1^ (PSMA), 0.67 fg mL^−1^ (hK2)
[80]	AFP, CEA, FER (**3**)	**St** (OPE0, OPE2, MBN)	AuNPs	MB	0.5 pg mL^−1^–500 pg mL^−1^ (AFP), 50 pg mL^−1^–2000 pg mL^−1^ (CEA), 10 pg mL^−1^–200 pg mL^−1^ (FER)	0.15 pg mL^−1^ (AFP), 20 pg mL^−1^ (CEA), 4 pg mL^−1^ (FER)
[91]	PSA, CEA, CA 19-9 (**3**)	**St** (MBA)	AuNPs	2D arrays of gold core−silver shell nanoparticles (Au@Ag core–shell NPs)	1 ng mL^−1^–1 pg mL^−1^ (PSA, CEA), 10–40 unit (U) mL^−1^ (CA19-9)	1 pg mL^−1^ (PSA, CEA), 10 unit (U) mL^−1^ (CA 19-9)
[83]	Glypican-1, EpCAMs), CD44V6 (**3**)	**St** (DTNB, MBA, TFMBA)	AuNPs	MB	0–2.3 × 10^8^ particles mL^−1^	2.3 × 10^6^ particles mL^−1^
[28]	IL-6, IL-8, IL-18 (**3**)	**St** (DTNB, FC, MBA)	AuNPs	Ag–Au substrate	0 ng mL^−1^–30 ng mL^−1^	2.3 pg mL^−1^, 6.5 pg mL^−1^, 4.2 pg mL^−1^ in a parallel, and 3.8 pg mL^−1^, 7.5 pg mL^−1^, 5.2 pg mL^−1^ in a simultaneous method for IL-6, IL-8 and IL-18, respectively
[95]	PSA, AFP, CEA, NSA (**4**)	**St** (MBA)	AuNPs	Gold substrate modified by Au–S bond (Au–SNBs)	1 ng mL^−1^–100 ng mL^−1^	10^−12^ mol mL^−1^
[81]	TNF-α, IFN-γ, IL-10, IL-6, IL-8 (**5**)	**St** (MBA, DTNB, TFMBA)	AuNPs with silver layers	MB	0 pg mL^−1^–10^5^ pg mL^−1^ (TNF-α)	4.5 pg mL^−1^ (TNF-α)

Explanation of abbreviations: 4-aminobphenyl (ABP), acetamide (AAD), 4-aminothiophenol (ATP), 1,4-benzenedithiol (1,4-BDT), cresyl violet (CV), dimercaptosuccinic acid (DMSA), **dual tag (Dt),** 5,5′-dithiobis(2-nitrobenzoic acid) (DTNB), 5,5′-dithiobis(succinimidyl-2-nitrobenzoate) (DSNB), fuchsin (FC), graphene oxide (GO), methylene blue (MB), mercaptobenzoic acid (MBA), 4-cyanobenzenethiol (MBN), methoxybenzenethiol (3-MeOBT), malachite green isothiocyanate (MGITC), 4-mercaptophenylboronic acid (MPBA), 2-methoxybenzenethiol (2-MeOBT), monoethylene glycol (MEG-OH), Nile blue A (NBA), 4-nitrobenzenethiol (NBT), 2-naphthalenethiol (2-NT), 4-nitrothiophenol (NTP), rhodamine 6G (R6G), **single tag (St),** toluidine blue (TB), 2,3,5,6-tetrafluoro-4-mercaptobenzoic acid (TFMBA), thionine (THI), triethylene glycol moiety (TEG–CO_2_H), tetramethylbenzidine (TMB), X-rhodamine-5-(and-6)-isothiocyanate (XRITC).

## Data Availability

Not applicable.

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
