# Peer review of "Recent Advances in Sandwich SERS Immunosensors for Cancer Detection"

_ijms, 2022, doi:10.3390/ijms23094740_

Round 1
Reviewer 1 Report
I have really enjoyed reading the review article from the authors. They carefully select paper in a predetermined time range, analyzing in detail all the advances in sandwich SERS immunosensors for cancer detection. My idea is that this review might be accepted after these revisions:
- Please specify which database search database was used for the analysis of literature reported in Figure 2.
- Some sentences require English rephrasing: "For many years, cancer has been one of the most popular diseases around the world"; "highly corelated"
- When reporting the literature examples, it would be appreciated, especially for the section 5 (Application of flow techniques) to include some figures extracted from papers in order to provide a tangible representation of the analytical performances to the readers.
Author Response
Answers to the Reviewer 1 (Ms. Ref. No.: ijms-1679093)
According to a kind letter encouraging us to resubmit our publication Recent advances in sandwich SERS immunosensors for cancer detection by A. Pollap and P. Świt in the International Journal of Molecular Sciences, I am sending the corrected manuscript. Please, find hereafter our answers to the reviewer’ comments, whom we would like to thank for the remarks.
Remarks of the referee have been respected in the following way.
Concerns of the Reviewer 1:
- Please specify which database search database was used for the analysis of literature reported in Figure 2.
Response: According to the reviewer's comment, we specified databases used for the analysis of literature.
“For this reason, it was decided to focus on the last ten years, starting from 2011, and it was decided to take into account the newest publications from 2022 based on Scopus, ScienceDirect, PubMed, Web of Science and Google Scholar databases.”
- Some sentences require English rephrasing: "For many years, cancer has been one of the most popular diseases around the world"; "highly corelated"
Response: We have made every effort to improve English, including consultations with native speakers. We have also been working on the manuscript to remove all linguistic errors.
- When reporting the literature examples, it would be appreciated, especially for the section 5 (Application of flow techniques) to include some figures extracted from papers in order to provide a tangible representation of the analytical performances to the readers.
Response: We are very grateful for your valuable suggestion. Unfortunately, it was impossible to obtain permission from the journals to place the source drawings in the required time for the revision. In order to provide a tangible representation of the analytical performances to the readers, we decide to prepare own figures for section 5 (Application of flow techniques) based on the cited articles. All figures represents the main ideas of microfluidic system, lab-on-a-chip platform and lateral flow bio-affinity assay combined with SERS detection for biomarker detection on the basis of antigen-antibody interactions.
Special thanks to the referee for the suggestions. Once again, we appreciate for Reviewer’ warm work earnestly and hope that the corrections will meet with approval.
Reviewer 2 Report
The review is well investigated but, English writing is not clear to understand.
The manuscript is too long to offer the decent perspective as a review paper.
In addition, most of Figure captions do not provide the details of each figure. Authors should provide the description of each figure.
Author Response
Answers to the Reviewer 2 (Ms. Ref. No.: ijms-1679093)
According to a kind letter encouraging us to resubmit our publication Recent advances in sandwich SERS immunosensors for cancer detection by A. Pollap and P. Świt in the International Journal of Molecular Sciences, I am sending the corrected manuscript. Please, find hereafter our answers to the reviewer’ comments, whom we would like to thank for the remarks.
Remarks of the referee have been respected in the following way.
Concerns of the Reviewer 2:
The review is well investigated but, English writing is not clear to understand.
Response: We have made every effort to improve English, including consultations with native speakers. We have also been working on the manuscript to remove all linguistic errors.
The manuscript is too long to offer the decent perspective as a review paper.
Response: Thank you very much for your valuable attention. We fully agree with you that the article is extensive. According to the sections presented in the article, our main goal was to provide the reader with the most valuable information as much as possible. We very carefully and thoroughly analysed the entire manuscript in terms of the possibility of shortening some parts. In each case, when we tried to remove certain fragments, the article became incomplete and did not give the opportunity to draw conclusions about the individual studies with adequate access to the essential details. Thank you very much for your valuable attention once again; we will most certainly follow your suggestions in future articles planned to be prepared.
In addition, most of Figure captions do not provide the details of each figure. Authors should provide the description of each figure.
Response: According to the reviewer suggestion, we provided a more detailed description of each figure in Figure captions.
“Figure 1. Exemplary applications of cancer biomarkers for clinical and medical purposes (based on [13,14]).”
“Figure 2. a) number of published articles during the period from 2011 to 2022 (February based on Scopus, ScienceDirect, PubMed, Web of Science and Google Scholar b) number of biomarker occurrences in the published article, where: AFP - α-fetoprotein, AFP-L3 - Lectin-reactive α-fetoprotein, ANG - angiogenin, B7-H6 - B7-H6 protein, BLV - bovine leukemia virus antigen gp51, CA 125 - carbohydrate antigen 125, CA 19-9 - carbohydrate antigen 19-9, CA15-3 - cancer antigen 15-3, CA27-29 - cancer antigen 27-29, CD19 - specific surface marker CD19, CD20 - specific surface marker CD20, CD44V6 - CD44 variant isoform 6, CEA - carcinoembryonic antigen, CK-19 - cytokeratin-19, CTCs - circulating tumor cells, EpCAMs - epithelial cell adhesion molecules, exosomes - tumor-derived exosomes, FER- ferritin, Glypican-1- glypican-1, hCE1- human carboxylesterase 1, HE4 - human epididymis protein 4, HER2 - human epidermal growth factor receptor 2, hK2 - human kallikrein 2, IFN-γ - interferon gamma, IL-10 - interleukin-10, IL-18 - interleukin-18, IL-6 - interleukin-6, IL-8 - interleukin-8, MMP-7 - matrix metalloproteinase-7, MMP-9 - matrix metalloproteinases-9, MP - metanephrine, MUC4 - mucin protein MUC4, NSE - neuron-specific enolase, OPN - osteopontin, p53 - protein p53, PSA - prostate specific antigen, PSMA - prostate-specific membrane antigen, Rac - adrenal stimulant ractopamine, SCCA - squamous cell carcinoma antigen, survivin, TNF-α - tumor necrosis factor α, VEGF - vascular endothelial growth factor.”
“Figure 3. Schematic construction of the sandwich SERS immunosensor in the multiplexed version. The most frequently used approach enabling single antigen analysis is shown in the red box.”
“Figure 4. Diagram showing the steps of sandwich-type SERS immunosensor developing: immunoprobe and substrate preparation and the immunosensor functioning.”
“Figure 5. Schematic representation of microfluidic system combined with SERS detection for biomarker detection based on antigen - antibody interactions (based on [87]).”
“Figure 6. Schematic representation of lab-on-a-chip platform combined with SERS detection for biomarker detection based on antigen - antibody interactions (based on [69]).”
“Figure 7. Schematic representation of lateral flow assay combined with SERS detection for biomarker detection based on antigen - antibody interactions (based on [68]).”
Special thanks to the referee for the suggestions. Once again, we appreciate for Reviewer’ warm work earnestly and hope that the corrections will meet with approval.